# Environmentally sensitive hotspots in the methylome of the early human embryo

Matt J Silver[1]*, Ayden Saffari[1], Noah J Kessler[2], Gririraj R Chandak[3], Caroline HD Fall[4], Prachand Issarapu[3], Akshay Dedaniya[3], Modupeh Betts[1], Sophie E Moore[1,5], Michael N Routledge[6,7], Zdenko Herceg[8], Cyrille Cuenin[8], Maria Derakhshan[1], Philip T James[1], David Monk[9,10], Andrew M Prentice[1]

[1]Medical Research Council Unit The Gambia at the London School of Hygiene and Tropical Medicine, Gambia, United Kingdom; [2]Department of Genetics, University of Cambridge, Cambridge, United Kingdom; [3]Genomic Research on Complex Diseases, CSIR-Centre for Cellular and Molecular Biology, Hyderabad, India; [4]MRC Lifecourse Epidemiology Unit, University of Southampton, Southampton General Hospital, Southampton, United Kingdom; [5]Department of Women and Children's Health, King's College London, London, United Kingdom; [6]School of Medicine, University of Leeds, Leeds, United Kingdom; [7]School of Food and Biological Engineering, Jiangsu University, Zhenjiang, China; [8]Epigenomics and Mechanisms Branch, International Agency For Research On Cancer, Lyon, France; [9]Biomedical Research Centre, University of East Anglia, Norwich, United Kingdom; [10]Bellvitge Institute for Biomedical Research, Barcelona, Spain

*For correspondence:
matt.silver@lshtm.ac.uk

Competing interest: The authors declare that no competing interests exist.

**Abstract** In humans, DNA methylation marks inherited from gametes are largely erased following fertilisation, prior to construction of the embryonic methylome. Exploiting a natural experiment of seasonal variation including changes in diet and nutritional status in rural Gambia, we analysed three datasets covering two independent child cohorts and identified 259 CpGs showing consistent associations between season of conception (SoC) and DNA methylation. SoC effects were most apparent in early infancy, with evidence of attenuation by mid-childhood. SoC-associated CpGs were enriched for metastable epialleles, parent-of-origin-specific methylation and germline differentially methylated regions, supporting a periconceptional environmental influence. Many SoC-associated CpGs overlapped enhancers or sites of active transcription in H1 embryonic stem cells and fetal tissues. Half were influenced but not determined by measured genetic variants that were independent of SoC. Environmental 'hotspots' providing a record of environmental influence at periconception constitute a valuable resource for investigating epigenetic mechanisms linking early exposures to lifelong health and disease.

## Editor's evaluation

This paper investigates the impact of seasonal variation (e.g. nutrition, environment, and infection) in rural subsistence farmer communities in the Gambia on DNA methylation levels in children. The authors identified a set of CpGs that are associated with season of conception and show that these associations are likely driven by periconceptional environmental influences. These findings open the door for future studies of environmentally sensitive CpGs to link early life exposures to diseases occurring later in life.

## Introduction

DNA methylation (DNAm) plays an important role in a diverse range of epigenetically regulated processes in mammals including cell differentiation, X-chromosome inactivation, genomic imprinting, and the silencing of transposable elements (TEs) (*Smith and Meissner, 2013*). DNAm can influence

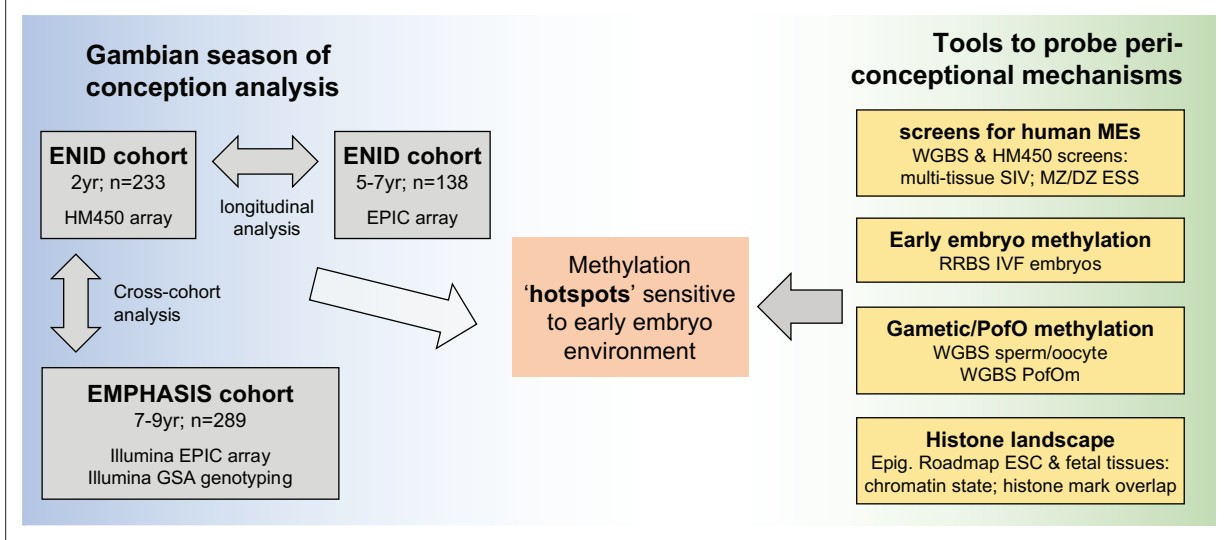

**Figure 1.** Study design. DNAm: DNA methylation; EPIC: Illumina Infinium MethylationEPIC BeadChip; Epig. Roadmap: Roadmap Epigenomics Consortium; ESC: embryonic stem cell; ESS: epigenetic supersimilarity; GSA: Global Screening Array; HM450: Illumina Infinium HumanMethylation450 BeadChip; IVF: in vitro fertilisation; MEs: metastable epialleles; MZ/DZ: monozygotic/dizygotic twins; PofO: parent of origin; RRBS: reduced representation bisulfite-seq; SIV: systemic interindividual variation; SoC: season of conception; WGBS: whole-genome bisulfite-seq. See **Tables 1 and 3** for further details of Gambian and public datasets used in this analysis.

gene expression and can in turn be influenced by molecular processes including differential action of methyltransferases and transcription factor (TF) binding (**Jeltsch, 2006**; **Feldmann et al., 2013**).

The human methylome is extensively remodelled in the very early embryo when parental gametic methylation marks are largely erased before acquisition of lineage and tissue-specific marks at implantation, gastrulation, and beyond (**Guo et al., 2014**). The days following conception may therefore offer a window of heightened sensitivity to external environmental influences, potentially stretching back to the period before conception coinciding with late maturation of oocytes and spermatozoa at loci that (partially) evade early embryonic reprogramming (**Fleming et al., 2018**).

The effects of early exposures on the mammalian methylome have been widely studied in animal models and periconceptional and early gestational factors including maternal folate and exposure to famine have been associated with DNAm changes in humans (**Gonseth et al., 2015**; **Tobi et al., 2015**; **Finer et al., 2016**). However, causal pathways are difficult to elucidate in human observational studies, and even randomised experimental designs are prone to confounding due to reverse causation from exposure-related effects (**Birney et al., 2016**).

Here, we address these limitations by exploiting a natural experiment in rural Gambia where conceptions occur against a background of repeating annual patterns of dry ('harvest') and rainy ('hungry') seasons with accompanying significant changes in energy balance, diet composition, nutrient status, and rates of infection (**Moore et al., 1999**; **Dominguez-Salas et al., 2013**). We assess the influence of seasonality on DNAm in two Gambian child cohorts (**Moore et al., 2012**; **Chandak et al., 2017**), one with longitudinal data, enabling robust identification of loci showing consistent effects at the ages of 2 years and in mid-childhood (**Figure 1**). Through prospective study designs we capture conceptions throughout the year and, in contrast to previous analyses in this population (**Waterland et al., 2010**; **Dominguez-Salas et al., 2014**; **Silver et al., 2015**), we use statistical models that make no prior assumptions about specific seasonal windows driving DNAm changes in offspring.

We probe potential connections between season of conception (SoC)-associated loci and early embryonic events by leveraging published data on loci with evidence for the establishment of variable methylation states in the early embryo that persist in post-gastrulation and postnatal tissues; namely loci demonstrating systemic interindividual variation (SIV) (**Kessler et al., 2018**; **Van Baak et al., 2018**) and/or epigenetic supersimilarity (ESS) (**Van Baak et al., 2018**; **Figure 1**). These loci bear the hallmarks of metastable epialleles (MEs), loci with methylation states that vary between individuals that were first identified in isogenic mice. MEs exhibit stable patterns of SIV indicating stochastic establishment of methylation marks prior to gastrulation when tissue differentiation begins

**Table 1.** Gambian seasonality-methylation analysis: cohort characteristics.

| Cohort | Sample size | Age | % male | Tissue | Methylation array |
|---|---|---|---|---|---|
| ENID (2 yr) | 233 | 2 years | 50.6 | Peripheral blood | Illumina Infinium HM450 |
| ENID (5–7 yr) | 138 | 5–7 years | 56.5 | | Illumina Infinium MethylationEPIC |
| EMPHASIS | 289 | 7–9 years | 54.3 | Peripheral blood | Illumina Infinium MethylationEPIC |

Note: ENID: Early Nutrition and Immune Development Trial (**Moore et al., 2012**); EMPHASIS: Epigenetic Mechanisms linking Pre-conceptional nutrition and Health Assessed in India and Sub-Saharan Africa (**Chandak et al., 2017**). Individuals with ENID longitudinal (5–7 yr) methylation data are a subset of those with methylation at 2 yr. There is no overlap between individuals included in the ENID and EMPHASIS cohorts.

(**Rakyan et al., 2002**), and several MEs have been shown to be sensitive to periconceptional nutrition in mice (**Anderson et al., 2012**). These loci thus serve as useful tools for studying the effects of early environment on DNAm by enabling the use of accessible tissues (such as blood) that can serve as a proxy for systemic (cross-tissue) methylation, and by pinpointing the window of exposure to the periconceptional period (**Gunasekara and Waterland, 2019**). We also investigate links with TEs and TFs associated with the establishment of methylation states in the early embryo, and assess the influence of genetic variation and gene-environment interactions. Finally, by comparing our results with public DNAm data obtained from sperm, oocytes, and multi-stage human embryos, we investigate links between SoC-associated loci, histone marks, gametic, and parent-of-origin-specific methylation (PofOm), and the establishment of DNAm states in early embryonic development.

Our identification of hotspots in the postnatal methylome that retain a record of environmental conditions during gametic maturation and/or in the very early embryo provides a valuable resource for the investigation of epigenetic mechanisms linking early-life nutritional and other exposures to lifelong health and disease.

## Results
### Identification of Gambian SoC-associated CpGs

Key characteristics of DNAm datasets from the two Gambian cohorts analysed in this study are provided in *Table 1*. DNAm differences associated with SoC are potentially confounded by season of sample collection effects in the ENID 2 yr dataset ($n$ = 233) since all samples were collected at age 24 months (*Figure 2A* top). This is not the case in the older EMPHASIS cohort ($n$ = 289; age 7–9 yr) where all samples were collected in the Gambian dry season (*Figure 2A* bottom). To account for the potential influence of season of collection effects, we therefore compared year-round DNAm signatures across ENID (2 yr) and EMPHASIS datasets by focussing on 391,814 autosomal CpGs ('array background') intersecting the Illumina HM450 and EPIC arrays used to measure DNAm in each dataset (*Table 2*). We modelled the effect of date of conception on DNAm using Fourier (or 'cosinor') regression (*Rayco-Solon et al., 2005*) which makes no prior assumptions about specific seasonal windows that might drive DNAm changes in offspring (see Materials and methods).

We began by identifying 768 SoC-associated CpGs showing significant seasonal variation in 2 year olds from the ENID cohort with a false discovery rate (FDR) < 5% (*Supplementary file 1a*; Materials and methods). Fourier regression models revealed a heterogeneous distribution of year-round methylation peaks and nadirs at these loci (*Figure 2B*). SoC-associated loci were highly enriched for loci exhibiting SIV/ESS previously identified in multi-tissue screens in adult Caucasians (*Kessler et al., 2018*; *Van Baak et al., 2018*), hereafter named 'MEs' for short (*Table 3*; *Figure 2B*, 26 ME CpGs marked in red; enrichment p = 2.5 × 10$^{-14}$). More than twice as many of these loci were within 100 bp of a putative ME ($n$ = 56; *Supplementary file 1a*). All identified loci showed increased seasonal amplitudes, defined as the distance between methylation peak and nadir, compared to matched and random controls (*Supplementary file 1b*; see *Table 2* and Materials and methods for justification and further details on selection of controls). Loci with the largest amplitudes tended to show increased methylation in conceptions in the Gambian rainy season (*Figure 2D* left) in line with our previous observations in this population (*Waterland et al., 2010*; *Dominguez-Salas et al., 2014*).

Next, we analysed SoC effects at these 768 loci in 7–9 year olds from the EMPHASIS cohort. Mean methylation at SoC-associated loci was strongly correlated across cohorts (*Appendix 1—figure 1*),

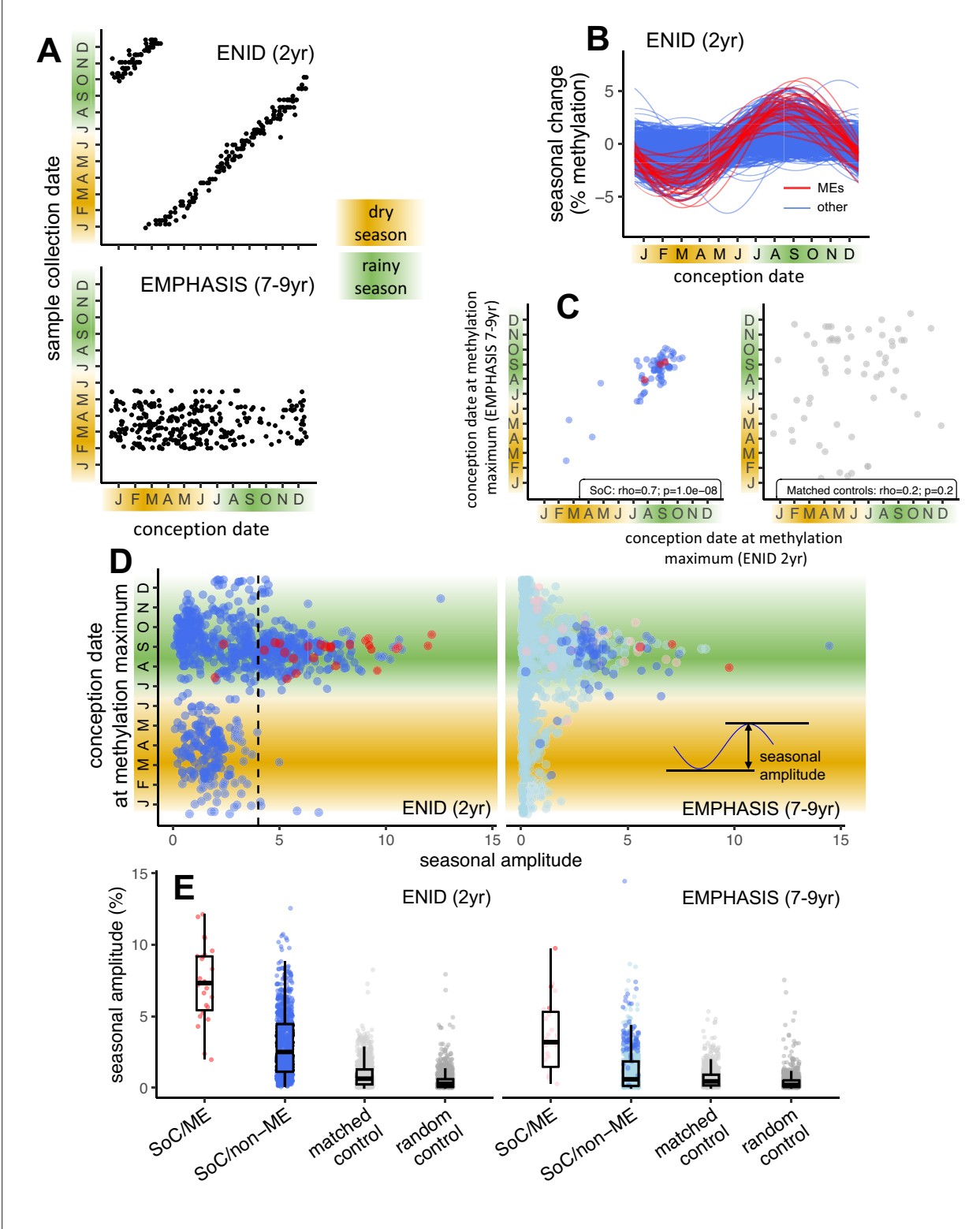

**Figure 2.** Identification of Gambian season of conception associated CpGs: ENID (2 yr) vs. EMPHASIS (7–9 yr) cross-cohort analysis. (**A**) Relationship between date of conception and date of sample collection for ENID (top) and EMPHASIS (bottom) cohorts. (**B**) Modelled seasonal change in methylation for 768 SoC-associated loci (false discovery rate [FDR] < 5%) in the ENID cohort. 26 ME CpGs are marked in red. (**C**) Conception date of modelled methylation maximum in each cohort for 61 CpGs significantly associated with SoC in both cohorts (left) and 61 matched controls (right). (**D**) (Left) Date of modelled DNAm maximum vs. seasonal amplitude in each cohort for 768 CpGs significantly associated with SoC in the ENID cohort. MEs

*Figure 2 continued on next page*

*Figure 2 continued*

are marked in red. Dashed line indicates SoC amplitude threshold used to identify SoC-CpGs. (Right) Same CpGs as left, but in the older EMPHASIS cohort. Significant SoC associations for this cohort are marked in a darker colour. (**E**) (Left) Seasonal amplitudes for SoC-associated CpGs that are (red) or are not (blue) MEs; along with amplitudes for 768 matched and random controls (light/dark grey respectively). (Right) As left but in the older EMPHASIS cohort. For EMPHASIS significant SoC associations are marked in a darker colour. Boxes represent the middle 50% of the data (inter-quartile range [IQR]); the line inside the box is the median, and whiskers represent values lying within 1.5 times the IQR.

and we found evidence of a similar effect of increased methylation in conceptions in the Gambian rainy season (*Figure 2D* right). Sixty-one loci (including three ME CpGs) were also significantly associated with SoC (FDR < 5%) in the older cohort (*Figure 2D* right). Notably, the date of conception at methylation maximum was highly correlated across these two independent and different aged cohorts (*Figure 2C* left; Spearman rho = 0.7, p = $1.0 \times 10^{-8}$). No significant correlation was observed at matched controls with similar methylation distributions to SoC-associated loci (*Figure 2C* right).

Given the strikingly similar seasonal patterns across the two cohorts, we next investigated reasons for the smaller number of SoC-associated loci (FDR < 5%) in the older (EMPHASIS) cohort. Focussing on the 768 SoC-associated loci in the ENID cohort, we found significant and relatively large effect size (SoC amplitude) decreases in the older cohort, with SoC effect attenuation most marked at loci that are also putative MEs (4.1% median methylation decrease, Wilcoxon p = $2.8 \times 10^{-6}$; non-ME CpGs: 1.9%, Wilcoxon p = $1.5 \times 10^{-77}$; *Figure 2E*; *Supplementary file 1c*). Corresponding SoC amplitude changes in matched and random controls were much smaller (0.2%, p = $2.0 \times 10^{-8}$).

Noting that loci with larger SoC amplitudes showed a more consistent pattern of seasonal variation, both within and between cohorts (*Figure 2C and D*), and reasoning that even transient molecular biomarkers of periconceptional environment in postnatal tissues could have biological significance, we focussed on 259 SoC 'hotspots' or SoC-CpGs with FDR < 5% and SoC amplitude ≥4% identified in the ENID 2 yr cohort (*Figure 2D* left, loci to the right of the dashed line; *Table 2*; *Supplementary file 1d*). Sensitivity analysis confirmed that SoC effect estimates at SoC-CpGs were robust to different modelling strategies with respect to estimated blood cell composition and batch effects (see Materials and methods).

We tested the hypothesis that SoC amplitudes at SoC-CpGs decrease with age by generating EPIC array methylation data on blood samples in a subset of *n* = 138 individuals from the ENID cohort at 5–7 yr (*Table 1*). This revealed a strongly consistent methylation signature at *n* = 157 replicating SoC-CpGs (*Figure 3A*; *Supplementary file 1d*; Spearman rho = 0.7, p < $2.2 \times 10^{-16}$). This analysis also provided further evidence of SoC effect attenuation with age at SoC-CpGs (*Figure 3B*; *Supplementary file 1d*; Wilcoxon signed rank test for difference in SoC amplitude p = $6.7 \times 10^{-12}$ for SoC-CpGs and p = 1.0 for matched controls). ENID longitudinal samples were collected in the rainy season (*Appendix 1—figure 2*), again exhibiting a different confounding structure with respect to season of sample collection compared to samples analysed at the earlier timepoint (ENID 2 yr) and in EMPHASIS (*Figure 2A*).

Critically, the date of conception at methylation maximum at SoC-CpGs was highly consistent across all three datasets analysed, with a distinct pattern of methylation maxima for conceptions falling within the August-September period, most markedly at putative MEs with independent evidence of

**Table 2.** CpG sets considered in this analysis.

| CpG set | Number of CpGs | Notes |
|---|---|---|
| Array background | 391,814 | Intersection of CpGs on Illumina HM450 (ENID 2 yr) and EPIC (EMPHASIS) cohort arrays, post QC |
| SoC-CpGs | 259 | SoC-associated CpGs with SoC effect size (SoC methylation amplitude) > 4% in the ENID 2 yr dataset |
| Matched controls | 259 | CpGs with similar methylation distributions to SoC-CpGs in the ENID 2 yr dataset* |
| Random controls | 259 | Random sample from array background |

*Matching methylation distributions determined by Kolmogorov-Smirnov tests (see Appendix 1—figure 16). QC: quality control; LRT: likelihood ratio test. See Materials and methods for further details.

**Table 3.** External datasets considered in this analysis.

| CpG set | Notes |
|---|---|
| Putative metastable epialleles (MEs) | 1881 ME/SIV/ESS CpGs overlapping array background identified in multi-tissue and MZ/DZ screens in *Van Baak et al., 2018* and *Kessler et al., 2018*. |
| Parent-of-origin-specific methylation (PofOm) | 699 Parent-of-origin-specific methylation loci identified in peripheral blood in *Zink et al., 2018*, overlapping array background. |
| Embryo DNAm data | RRBS data for inner cell mass and embryonic liver (<10 weeks' gestation) from *Guo et al., 2014*. |
| Sperm DNAm data | WGBS data from *Okae et al., 2014*. |
| Germline DMRs (gDMRs) | Regions differentially methylated in sperm and oocytes identified in WGBS data by *Sanchez-Delgado et al., 2016*. |
| Transposons (ERVs) | ERVs determined by RepeatMasker were downloaded from the UCSC h19 annotations repository. |
| Transcription factor ChIP-seq | ZFP57, TRIM28, and CTCF transcription factor binding sites identified from ChIP-seq in human embryonic kidney and hESCs are described in *Kessler et al., 2018*. |
| Chromatin state predictions and histone three marks | Chromatin state predictions for H1 ESCs, fetal brain, fetal muscle, and fetal small intestine generated using *Ernst and Kellis, 2012*, from *Roadmap Epigenomics Consortium et al., 2015*. Histone mark data are from the same source. |

ME: metastable epiallele. SIV: systemic interindividual variation. ESS: epigenetic supersimilarity. MZ/DZ: monozygotic/dizygotic twins. PofOm: parent-of-origin methylation. RRBS: reduced representation bisulfite-seq. DMR: differentially methylated region. ERV: endogenous retrovirus. ESCs: embryonic stem cells. See materials and methods for further details.

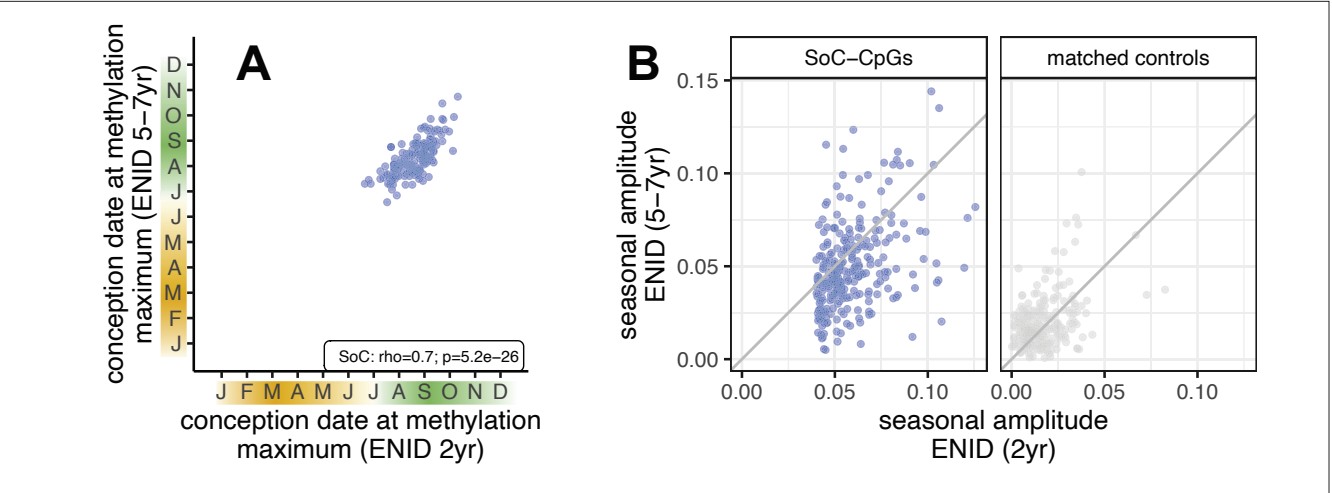

**Figure 3.** Identification of Gambian season of conception associated CpGs: ENID 2 yr vs. 5–7 yr longitudinal analysis. (**A**) Relationship between conception date of modelled methylation maximum measured at 2 and 5–7 yr in the same $n = 138$ individuals from the ENID cohort. $n = 157$ SoC-CpGs with a significant SoC association (false discovery rate [FDR] < 5%) at 5–7 yr are plotted. (**B**) Change in seasonal amplitude between 2 and 5–7 yr for $n = 259$ SoC-CpGs (left) and matched controls (right) with DNAm measured in the same $n = 138$ individuals from the ENID cohort. There is evidence of SoC effect attenuation with age at SoC-CpGs, but not at matched controls (Wilcoxon signed rank test for difference in SoC amplitude $p = 6.7 \times 10^{-12}$ and $p = 1.0$, respectively).

establishment in the early embryo (**Figure 4A**). The August to September period corresponds to the peak of the Gambian rainy season, a strong validation of our previous studies in babies and infants that focussed on conceptions at peak seasons only, with similar observations of higher methylation in conceptions at the peak of the Gambian rainy season compared to peak dry season (**Waterland et al., 2010**; **Dominguez-Salas et al., 2014**; **Silver et al., 2015**; **Van Baak et al., 2018**). Methylation minima fall within the February-April period, corresponding to the peak of the dry season (**Appendix 1—figure 3**).

Our observation of a remarkably similar season of conception signature across two cohorts and three datasets with different confounding structures with respect to season of sample collection, batch, and biological variables (**Supplementary file 1p-r**), combined with evidence from cross-cohort and longitudinal analyses of SoC effect attenuation with age led us to conclude that SoC-CpGs act as robust sentinels of SoC-associated effects persisting at least until the age of 2 years.

## Properties of SoC-CpGs

SoC-CpGs are distributed throughout the genome and cluster together in several regions (**Appendix 1—figure 4**). Noting that the number of clusters is relatively insensitive to the inter-CpG distance used to define them (**Appendix 1—figure 5**), we identified 56 distinct SoC-CpG clusters and 105 'singletons' (SoC-CpGs with no close neighbours) using a maximum inter-CpG distance of 5 kbp (**Supplementary file 1e**). With this definition, 59% of SoC-CpGs fell within clusters (**Supplementary file 1f**). Of note, SoC effect amplitudes and cross-cohort correlations were greater at SoC-CpGs falling within clusters than with singletons (**Appendix 1—figure 6**).

Several SoC-CpG clusters extend over more than 500 bp, notably a cluster mapping to *IGF1R* which spans 872 bp and covers 7 CpGs (**Supplementary file 1e**, **Appendix 1—figure 7**). All but one of these 7 CpGs were also significantly associated with SoC (FDR < 5%) in the older EMPHASIS cohort (**Supplementary file 1d**).

Compared to array background, SoC-CpGs are intermediately methylated, most notably at putative MEs (**Figure 4B**), and they tend to fall within CpG islands compared to array background and matched controls (**Figure 4C**). SoC-CpGs are also highly enriched for MEs (21-fold enrichment, $p = 3.0 \times 10^{-23}$, cluster-adjusted: 17-fold, $p = 3.1 \times 10^{-11}$; **Supplementary file 1g**; see Materials and methods for further details on cluster-based adjustments). The number of SoC-CpGs directly overlapping previously identified MEs is small ($n = 24$), although 49 SoC-CpGs (19%) fall within 100 bp of an ME (**Supplementary file 1d**). Further investigation revealed that a large majority ($n = 19/24$) of

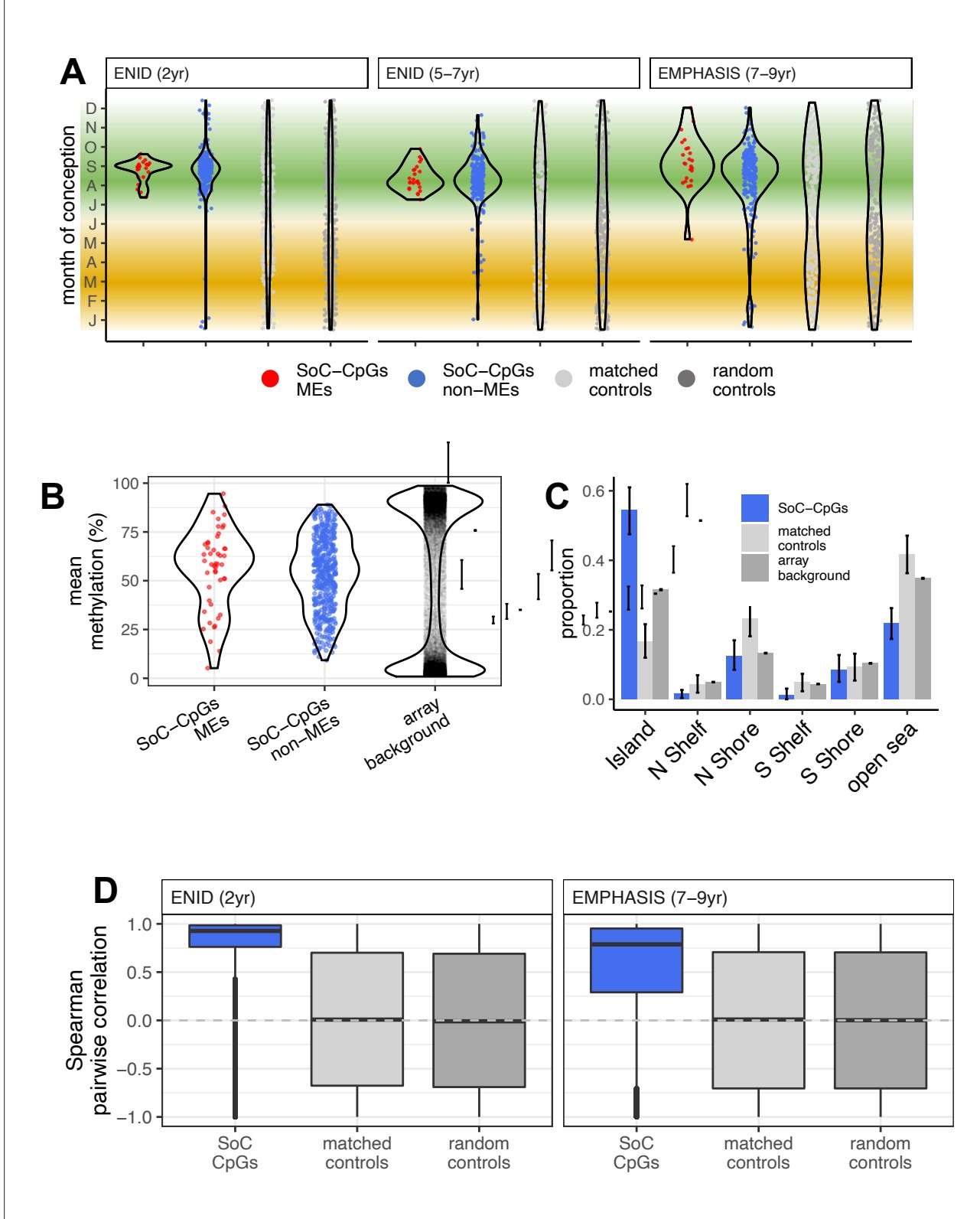

**Figure 4.** Properties of SoC-CpGs. (**A**) Date of conception at modelled methylation maxima for 259 SoC-CpGs and 259 corresponding matched and random controls across all three analysed datasets. Green and yellow bands indicate the extent of the rainy and dry seasons, respectively. (**B**) SoC-CpGs show increased intermediate methylation compared to array background (data from ENID 2 yr and EMPHASIS cohorts combined). (**C**) Distribution of 259 SoC-CpGs [including MEs], 259 matched controls and array background with respect to CpG islands. Error bars are bootstrapped 95% CIs. N/S

*Figure 4 continued on next page*

*Figure 4 continued*

Shore/Shelf: North/South Shore/Shelf, respectively (regions proximal to CpG Islands defined in Illumina manifest). (**D**) Distribution of pairwise Spearman correlations for CpG sets in the ENID (left) and EMPHASIS (right) cohorts. Boxplot elements as described in *Figure 2E*.

overlapping MEs were identified in our previous screen for SIV, with a smaller number exhibiting ESS (*n* = 7/24; *Supplementary file 1g*). No MEs overlap methylation distribution-matched controls.

Pairwise methylation states are highly correlated at a large majority of SoC-CpGs in both cohorts, so that the same individuals tend to have relatively high or low methylation at multiple SoC-CpGs (*Figure 4D*). Pairwise correlations are not driven by increased correlation within SoC-CpG clusters (*Appendix 1—figure 8*), and methylation at matched and random controls is uncorrelated, thus reducing the possibility that these correlations are driven by statistical artefacts.

Finally, SoC-CpG reliability is classified as 'excellent' (median ICC = 0.76) using probe reliability estimates from a recent repeated measures study (*Sugden et al., 2020*). Reliability of matched control CpGs is classified as 'good' (median ICC = 0.68) using the same method.

## Early stage embryo, gametic, and PofOm

Given the strong enrichment for MEs within the set of SoC-CpGs, we next analysed links to methylation changes in early stage human embryos, as we have done previously for putative MEs identified in a whole-genome bisulfite-seq (WGBS) multi-tissue screen (*Kessler et al., 2018*). We aligned our data with public reduced representation bisulfite-seq (RRBS) data from human IVF (in vitro fertilisation) embryos (*Guo et al., 2014*) and obtained informative methylation calls for 112,380 array background CpGs covered at ≥10× read depth in inner cell mass (ICM, pre-gastrulation) and/or embryonic liver (post-gastrulation) tissues. As previously noted at putative MEs (*Kessler et al., 2018*), we found a distinctive pattern of increased incidence of intermediate methylation states at SoC-CpGs in post-gastrulation embryonic liver tissue, strongly contrasting with a general trend of genome-wide hyper- and hypomethylation at loci mapping to array background (*Figure 5A*). A similar pattern of increased incidence of intermediate methylation states was observed at distribution-matched controls.

We previously observed consistent hypomethylation at MEs across all gametic and early embryonic developmental stages, most notably in sperm (*Kessler et al., 2018*). We tested the latter observation at SoC-CpGs by aligning our data with public sperm WGBS data (*Okae et al., 2014*), restricting our analysis to 294,240 CpGs mapping to array background that were covered at ≥10×. SoC-CpGs tended to be hypomethylated in sperm, compared to loci mapping to matched control CpGs and array background, respectively (*Figure 5B and C*). Intermediate methylation states at SoC-CpGs were preserved in both Gambian cohorts irrespective of sperm methylation states, in contrast to array background CpGs where methylation distributions strongly reflected sperm hypomethylation status (ENID cohort: *Figure 4D* left; EMPHASIS cohort: *Appendix 1—figure 9* left).

Our observation of an increased incidence of sperm hypomethylation at SoC-associated loci, together with existing evidence that imprinted genes may be sensitive to prenatal exposures (*Silver et al., 2015*; *Monk et al., 2019*; *James et al., 2018b*), prompted us to investigate a potential link between SoC-CpGs and PofOm. A recent study used phased WGBS methylomes to identify regions of PofOm in 200 Icelanders (*Zink et al., 2018*). We analysed 699 of these PofOm CpGs overlapping array background (*Table 3*) and observed strong enrichment for PofOm CpGs at SoC-CpGs and at all MEs on the array (18- and 15-fold enrichment, p = 3.0 × 10⁻⁸ and 1.8 × 10⁻³⁶, respectively; *Supplementary file 1h*; *Figure 5D* right, PofOm CpGs marked as green triangles). No enrichment was observed at distribution-matched controls (*Supplementary file 1h*). PofOm enrichment at SoC-CpGs is driven by a large PofOm region spanning 6 CpGs on chr15 at *IGF1R* (*Supplementary file 1d*); along with two singleton PofOm SoC-CpGs, one on chr18 close to *PARD6G* and the other in the Prader-Willi syndrome-associated imprinted region neighbouring *MAGEL2*, also on chr15. All of these loci have increased methylation in the rainy season, with SoC effect sizes (methylation amplitudes) ranging from 4.1% to 8.4% (median 6.1%; *Supplementary file 1d*).

Regions of PofOm detected in postnatal samples tend to be differentially methylated in gametes (*Zink et al., 2018*), and may thus have evaded epigenetic reprogramming in the pre-implantation embryo (*Monk et al., 2019*). We tested this directly by interrogating data from a whole-genome screen for germline differentially methylated regions (gDMRs) that persist to the blastocyst stage and beyond (*Sanchez-Delgado et al., 2016*). In this analysis, gDMRs were defined as contiguous 25 CpG

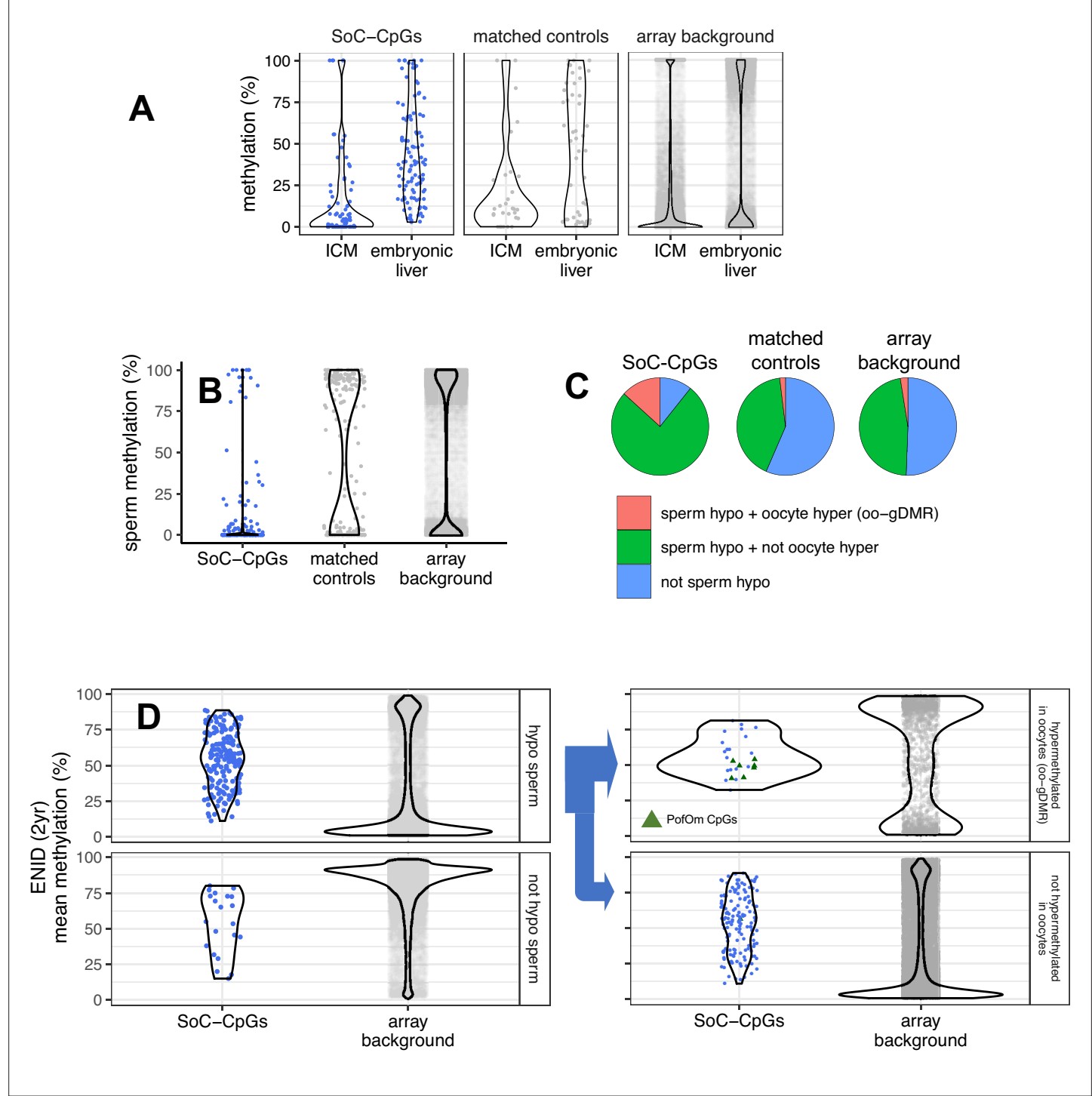

**Figure 5.** Early stage embryo, gametic, and parent-of-origin-specific methylation (PofOm). (**A**) Methylation distribution of SoC-CpGs, matched controls, and array background in pre-gastrulation inner cell mass (ICM) and post-gastrulation embryonic liver, measured in reduced representation bisulfite-seq (RRBS) embryo methylation data from *Guo et al., 2014*. Data comprises 112,380 CpGs covered at ≥10× in ICM and/or embryonic liver that overlap array background, including 118 SoC-CpGs and 51 matched controls. (**B**) Methylation distribution of SoC-CpGs, matched controls, and array background in sperm whole-genome bisulfite-seq (WGBS) data from *Okae et al., 2014*. Data comprises 294,240 CpGs covered at ≥10× including 196 SoC-CpGs and 207 matched controls. (**C**) Sperm methylation and oocyte germline differentially methylated region (oo-gDMR) status at 196 SoC-CpGs covered at ≥10× in Okae et al. sperm WGBS data. Sperm hypomethylation is defined as methylation ≤25%. oo-gDMRs defined as sperm methylation <25% and oocyte methylation >75% in WGBS analysis by *Sanchez-Delgado et al., 2016*. (**D**) Mean methylation at SoC-CpGs and array background measured in *n* = 233 individuals in the ENID (2 yr) cohort stratified by sperm and oocyte methylation status. Left: Methylation stratified according to sperm hypomethylation

*Figure 5 continued on next page*

*Figure 5 continued*
status (*n* = 175 SoC-CpGs hypomethylated in sperm, *n* = 21 not hypomethylated). Right: As left but with loci hypomethylated in sperm further stratified according to oo-gDMR status (*n* = 26 SoC-CpGs hypermethylated in oocytes/oo-gDMR, *n* = 149 not hypermethylated). Sperm/oo-gDMR status thresholds as for *Figure 4C*. Eight PofOm CpGs (green triangles) are those identified in *Zink et al., 2018*.

regions that were hypomethylated (mean DNAm < 25%) in one gamete and hypermethylated (mean DNAm > 75%) in the other. We began by observing strong enrichment for oocyte (maternally methylated), but not sperm gDMRs, at all PofOm loci identified by *Zink et al., 2018* (*Supplementary file 1h*), confirming previous observations of an excess of PofOm loci that are methylated in oocytes only (*Zink et al., 2018*). This enrichment was particularly strong for oocyte gDMRs (oo-gDMRs) persisting in placenta (*Supplementary file 1h*). We next analysed SoC-CpGs and MEs and again found evidence for strong enrichment of oocyte, but not sperm gDMRs at these loci (6.2-fold oo-gDMR enrichment, p = 2.3 × 10$^{-16}$ at SoC-CpGs; 2.9-fold, p = 1.2 × 10$^{-24}$ at MEs), including after adjustment for CpG clustering (*Supplementary file 1h*). Of note, 14% (36/259) of SoC-CpGs overlapped oo-gDMRs, in strong contrast to matched and random controls (*Figure 5C*). These clustered into 19 distinct oo-gDMR regions (*Supplementary file 1i*) – more than six times the number identified as exhibiting PofOm by Zink et al. (three regions; *Supplementary file 1d*).

A large majority of SoC-CpGs that are hypomethylated in sperm are not oo-gDMRs (i.e. they are not hypermethylated in oocytes) (*Figure 5C and 5D* bottom right), suggesting that factors associated with regional sperm hypomethylation rather than differential gametic methylation may be a key driver of sensitivity to periconceptional environment at these loci.

## SoC-CpG overlap with predicted chromatin states

We assessed the overlap of SoC-CpGs with predicted chromatin states generated from histone marks in various cell lines and tissues by the *Roadmap Epigenomics Consortium et al., 2015*. Given our interest in methylation states associated with periconceptional environment that persist into early postnatal life, we focussed on data from H1 embryonic stem cells (ESCs) and fetal tissues (fetal brain, muscle, and small intestine) derived from all three germ layers, described as having the 'highest quality' epigenomes (see Figure 2 in *Roadmap Epigenomics Consortium et al., 2015*). Around half of all SoC-CpGs overlapped sites with predicted transcriptional or regulatory function, with relatively

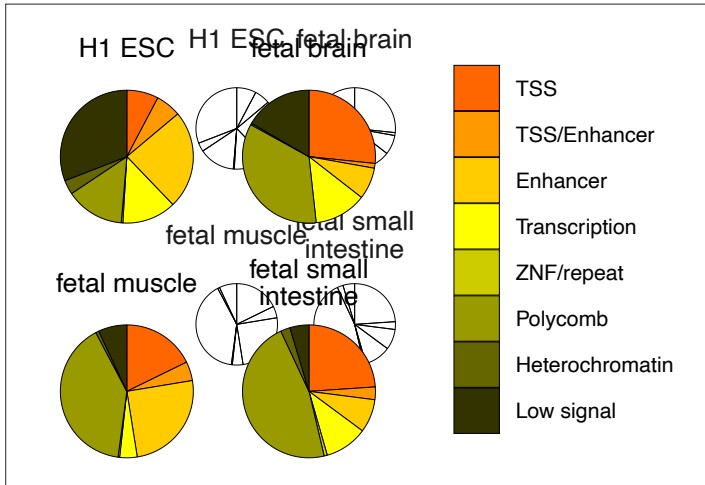

**Figure 6.** SoC-CpG overlap with predicted chromatin states. Chromatin states predicted by ChromHMM (*Ernst and Kellis, 2012*) from chromatin marks in four cell lines and tissues generated by the *Roadmap Epigenomics Consortium et al., 2015*. Predicted states for all 259 SoC-CpGs are shown. Predictions from the ChromHMM 15-state model are collapsed to eight states for clarity. TSS: active transcription start site/flanking active TSS/ bivalent or poisedTSS; TSS/enhancer: flanking bivalent TSS/enhancer; enhancer: enhancer/bivalent enhancer/ genic enhancer; transcription: strong/weak transcription/transcription at gene 5' and 3'; ZNF/repeat: zinc finger genes and repeats; polycomb: repressed/weak repressed polycomb; heterochromatin; low signal: low signal in all marks states used as inputs to ChromHMM.

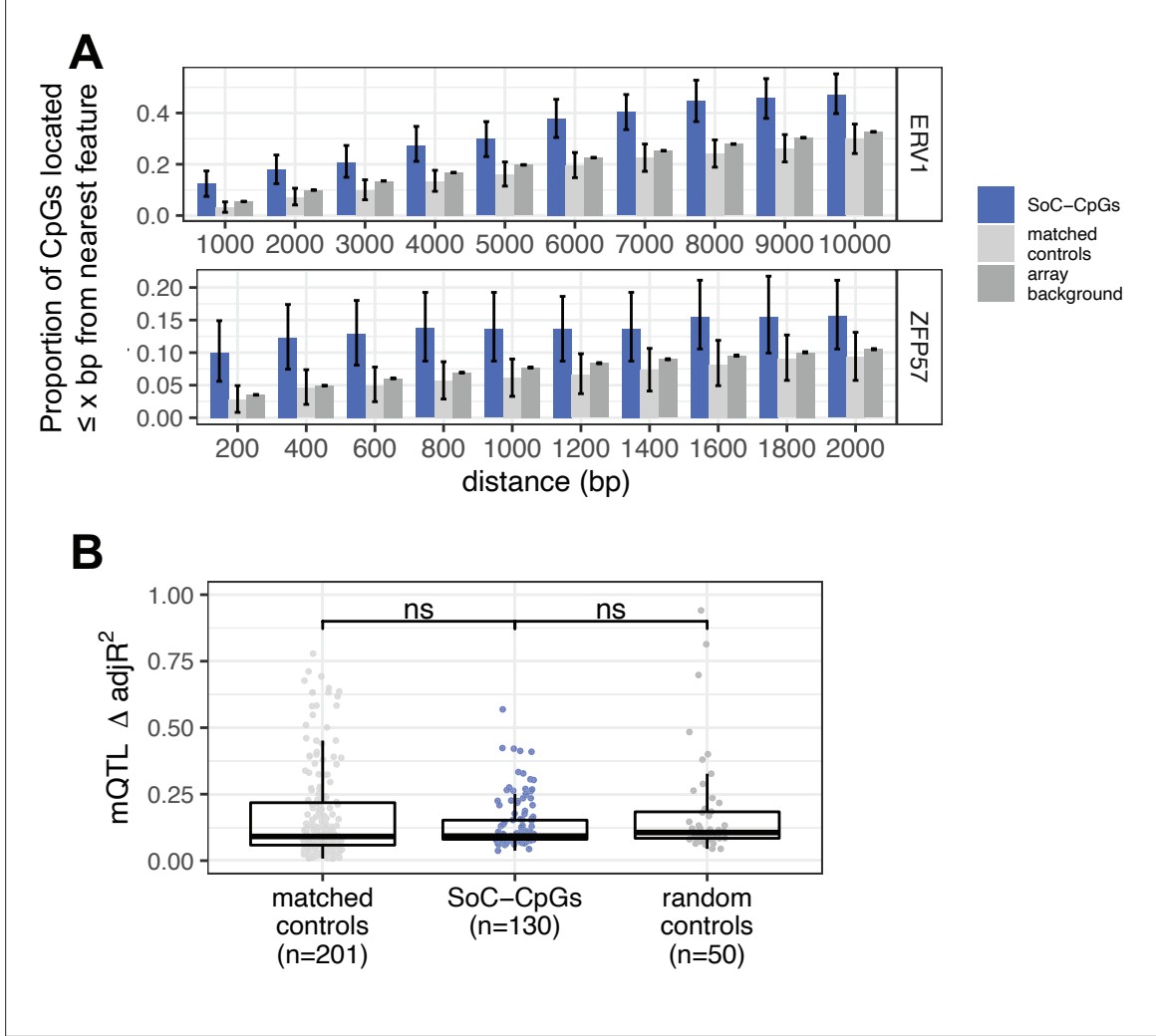

**Figure 7.** Links between endogenous retroviruses (ERVs), ZFP57 binding sites, genetic variation, and DNAm at season of conception associated loci. (**A**) Proportion of SoC-CpGs, matched controls, and array background CpGs proximal to ERV1 endogenous retroviral elements (top) and ZFP57 binding sites (bottom), within the specified distance. CpG clustering effects are removed by sampling a single CpG from each cluster (see Materials and methods). Error bars are bootstrapped 95% CIs. (**B**) Proportion of methylation variance explained by methylation quantitative trait locus (mQTL) for matched controls, SoC-CpGs and random controls. Only CpGs with at least one significant mQTL are plotted (n = 201, 130, and 50 respectively; see Materials and methods for further details). Boxplot elements as described in **Figure 2E**.

few overlapping constitutive heterochromatic regions (**Figure 6**). Overlaps with specific histone marks in H1 ESCs are given in **Appendix 1—figure 10**. As expected, given predicted chromatin states, many show a predominance of overlapping H3K4me1 and H3K27me3 marks and combinations thereof, suggestive of active or poised enhancers.

## Enrichment of transposable elements and transcription factors associated with genomic imprinting

Variable methylation states at MEs are associated with TEs in murine models (**Waterland and Jirtle, 2003**; **Kazachenka et al., 2018**), and we have previously observed enrichment for proximity to two classes of endogenous retroviruses, ERV1 and ERVK, at putative human MEs (**Silver et al., 2015**; **Kessler et al., 2018**). Here, we found evidence that SoC-CpGs are enriched for proximity to ERV1 (**Figure 7A** top) but not ERVK retroviral elements (**Appendix 1—figure 11**; **Table 3**).

Enrichment for PofOm and gDMRs at SoC-CpGs suggests a potential link to mechanisms implicated in the maintenance of PofOm and genomic imprinting in the early embryo. Our previous analysis of MEs identified from WGBS data found enrichment for proximal binding sites for three TFs (CTCF,

**Table 4.** Methylation quantitative trait loci (mQTL) associated with SoC-CpGs and controls.

| CpG set | Number of CpGs with mQTL | Number of mQTL (*cis/trans*) | Median number of mQTL per CpG (IQR) | Methylation variance explained* |
|---|---|---|---|---|
| SoC-CpGs | 130 (50%) | 2771 (2549/222) | 6 (2–30) | 0.09 (0.08–0.15) |
| Matched controls | 201 (78%) | 7886 (7417/469) | 15 (4–50) | 0.09 (0.06–0.21) |
| Random controls | 50 (19%) | 1512 (1476/36) | 7 (2–35) | 0.1 (0.08–0.18) |

*delta adjusted $R^2$ (see Materials and methods); IQR: inter-quartile range.

ZFP57, and TRIM28) identified through ChIP-seq of embryonic stem and kidney cells that are linked to maintenance of PofOm at imprints (*Kessler et al., 2018*). Here, we found evidence for enrichment of proximal ZFP57 binding sites within 2 kbp of a SoC-CpG as previously observed at MEs, but we found no evidence for enrichment of proximal CTCF or TRIM28 binding sites in this array-based study (*Figure 7A* bottom; *Appendix 1—figure 11*; *Table 3*).

## Influence of genotype

Genetic variation is a major driver of interindividual variation in DNAm via methylation quantitative trait loci (mQTL) (*Gaunt et al., 2016*). We explored the influence of mQTL on SoC-CpGs in the EMPHASIS cohort for which we had genotype data on 284 individuals measured at >2.6M SNPs after imputation from the Illumina Global Screening Array (GSA) and subsequent LD pruning (see Materials and methods). The majority of mQTL effects occur in *cis* (*Gaunt et al., 2016*). In order to maximise power, we therefore adopted a two-step approach where we performed separate screens for mQTL in *cis* (defined as SNPs within 1 Mb of an associated CpG; *Gaunt et al., 2016*) and *trans* (all others), and compared our findings at SoC-CpGs with matched and random control CpGs (Materials and methods). Half of SoC-CpGs had one or more associated mQTL compared with 78% and 19% of matched and random controls, respectively; 92% of SoC-CpG mQTL were in *cis* (*Table 4*).

We next compared methylation variance explained by significant mQTL using adjusted $R^2$ values for all SoC-CpGs and controls with at least one genome-wide significant mQTL (FDR < 5%; *n* = 130, 201, and 50 CpGs for SoC-CpGs, matched and random controls, respectively; *Table 4*). These values were compared to a baseline model that included the same set of covariates (principal components [PCs], age and sex) used in Fourier regression models for the main seasonality analysis, in order to account for potential differences in additional variance explained by other covariates and unmeasured factors (see Materials and methods). There was no difference in additional variance explained by significant mQTL between SoC-CpGs and both sets of control CpGs (*Figure 7B*; *Table 4*).

To assess the potential for genetic confounding of SoC-associated DNAm signals at SoC-CpGs with associated mQTL, we tested all SoC-CpG-mQTL for association with season of conception using an allelic model. After accounting for multiple testing, no significant SoC-mQTL associations were identified (*Supplementary file 1j* and Materials and methods). Our observations that (i) SoC-CpGs are distributed throughout the genome; (ii) SoC-CpG mQTL occur primarily in *cis*; and (iii) none are associated with SoC; strongly suggest that SoC-methylation associations at SoC-CpGs are not confounded by genetic variation.

Finally, we searched for GxE (SoC) effects, again performing separate tests for SNPs in *cis* and *trans*. No GxE associations were identified after correcting for multiple testing (see Materials and methods).

## Influence of genetic ancestry

Eighty percent of the population of the Kiang West region of The Gambia from which the cohorts analysed in this study are drawn are of Mandinka ethnicity, with the majority of the remainder Fula (*Hennig et al., 2017*). This is evident from a genome-wide principal component analysis (PCA) of genetic variation in the EMPHASIS cohort, where we observed a distinct cluster of 16 individuals from a single village which is predominantly Fula (*Appendix 1—figure 12*). Individuals in the main ENID cohort were drawn from the same Kiang West villages as the EMPHASIS study, but we were unable to directly adjust for potential confounding effects due to genetic ancestry since no genetic data

was available for this cohort. Based upon the EMPHASIS cohort PCA and our knowledge of village population structures, we reasoned that village of origin is a useful proxy for genetic ancestry in the ENID cohort and performed a sensitivity analysis with an additional covariate dichotomised according to whether an individual came from the predominantly Fula village. The first two genetic PCs were used as adjustment covariates for the corresponding EMPHASIS analysis. Results from this ethnicity-adjusted sensitivity analysis were not materially different from those obtained for the main analysis (*Appendix 1—figure 13*; *Supplementary file 1s*).

## Overlap of SoC-CpGs with existing studies

To place our findings in the context of existing literature on associations between DNAm and nutrition-related exposures, including exposure to famine conditions, folate supplementation in gestation and previous reported associations with Gambian SoC, we checked for overlaps between SoC-CpGs and loci identified in a recent review by *James et al., 2018b*. Many cited studies including the majority of previous work in The Gambia used pyrosequencing and other methylation platforms targeting loci not covered by Illumina arrays. However, a total of 57 previously identified loci did overlap or partially overlap array background. None of these overlapped a SoC-CpG within 1 kbp. We also checked for overlaps with the larger set of SoC-associated CpGs not passing the 4% minimum effect size threshold, and found a single CpG (cg17434309) mapping to *IGF2* that was within 1 kbp of two previously identified loci, one linking maternal plasma vitamin B12 with cord blood methylation (*Ba et al., 2011*), and the second linking gestational famine to blood methylation in older adults (*Tobi et al., 2012*).

We next looked for overlaps between SoC-CpGs and CpGs identified in the epigenome-wide association studies (EWAS) Catalog (http://ewascatalog.org/), a manually curated database of significant results ($p < 1 \times 10^{-4}$) from published EWAS. This search produced published associations for 167 out of the 259 SoC-CpGs, mapping to 27 unique traits covering a range of pre- and postnatal exposures (*Supplementary file 1l*). Noteworthy amongst the most frequently reported associations with SoC-CpGs (*Supplementary file 1m*) in the context of our study were those with sex, gestational age, maternal smoking in pregnancy, maternal plasma folate levels, and adult body mass index (BMI) with 109, 45, 16, 6, and 1 associated SoC-CpG(s), respectively.

We investigated some of these links using data from the 2 yr ENID cohort considered in our main analysis and confirmed multiple significant associations with infant sex but not gestational age or maternal folate at conception (*Supplementary file 1n*). Links with adult BMI and maternal smoking were not considered as adult BMI was not available and the incidence of smoking is extremely low in our study population.

All Fourier regression models in our main SoC analysis included sex as an adjustment covariate. The finding that multiple SoC-CpGs were associated with sex in the EWAS Catalog, with replication of this association in the ENID cohort, was therefore surprising. This prompted us to check for a residual confounding effect due to sex by repeating our analysis with methylation values pre-adjusted for sex using a regression model with sex as the only adjustment covariate, prior to running the full regression models. This produced near identical results to the main analysis without pre-adjustment. This, combined with our observation that sex is not associated with any tested batch or biological covariates (*Supplementary file 1p-r*), strongly suggests that the observed SoC associations were not driven by confounding due to sex.

Finally we searched for SoC-CpGs within 1 kbp of SNP associations ($p < 1 \times 10^{-5}$) in the GWAS Catalog (*Buniello et al., 2019*), since DNAm could mediate GWAS signals in genomic regions where functional effects are difficult to elucidate (*Do et al., 2017*). Eleven SoC-CpGs mapped to a total of 12 SNPs associated with eight unique traits in the GWAS Catalog (*Supplementary file 1o*). Notable traits from a developmental programming perspective were those linked to childhood obesity (*Comuzzie et al., 2012*) (2 SoC-CpGs) and QRS traits associated with cardiovascular mortality in adults (1 SoC-CpG) (*Evans et al., 2016*; *van der Harst et al., 2016*).

## Discussion

We have utilised a natural, seasonal experiment in rural Gambia whereby human conceptions are 'randomised' to contrasting environmental (especially dietary) conditions to examine whether this

differential exposure leaves a discernible signature on the offspring methylome. We analysed methylation data from two independent, different-aged cohorts and identified 259 'SoC-CpGs' with evidence of sensitivity to SoC in infancy. We found evidence of SoC effect attenuation with age, but SoC-related temporal patterns were nonetheless strikingly similar, suggestive of a common effect of periconceptional environment. Importantly, the analysed datasets have contrasting confounding structures, notably with regard to the timing of sample collection, the latter eliminating potential confounding due to seasonal differences in leukocyte composition. These results, derived from analysis of Illumina array data with limited coverage, suggest there may be many more hotspots sensitive to the periconceptional environment across the human methylome.

This analysis builds on previous epigenetic studies in this setting that have focussed on single cohorts and analysed methylation differences between individuals conceived at the peaks of the Gambian dry and rainy seasons only (*Waterland et al., 2010*; *Dominguez-Salas et al., 2014*; *Silver et al., 2015*; *Van Baak et al., 2018*; *Kühnen et al., 2016*). Increased methylation in offspring conceived at the peak of the Gambian rainy season is consistent with previous findings and this observation is now greatly strengthened by the application of Fourier regression to model year-round conceptions, an approach that makes no prior assumption of when methylation peaks and nadirs may occur. The number of identified SoC-CpGs is also substantially increased in this study. Comparisons with array background and control CpGs matching SoC-CpG methylation distributions increase confidence that these findings are not statistical artefacts.

Triangulation with other public data provides multiple lines of evidence supporting the notion that methylation states at these loci are established in the periconceptional period. First, they are highly enriched for putative MEs and related loci identified in other studies with characteristic methylation signatures suggestive of establishment early in embryonic development (*Kessler et al., 2018*; *Van Baak et al., 2018*). Second, like MEs, season-associated loci exhibit unusual methylation dynamics in early stage embryos (*Kessler et al., 2018*). Third, they have distinctive gametic methylation patterns, notably hypomethylation in sperm (in common with putative MEs; *Kessler et al., 2018*), and differential gametic and PofOm in a subset. Increased incidence of hypomethylated states in sperm at SoC-CpGs may reflect their enrichment at CpG islands (*Molaro et al., 2011*), sequence features that are largely refractory to protamine exchange, with the possibility for retaining epigenetic function associated with histone modifications into the early embryo (*Hammoud et al., 2009*). Fourth, many overlap H3K4me1 and H3K27me3 marks which coordinate transient gene expression and early lineage commitment in human ESCs, in part through demarking poised enhancers (*Rada-Iglesias et al., 2011*).

A large majority of SoC-CpGs have not previously been identified as MEs, but given the supporting evidence described above, we speculate that many are likely to be so. Indeed, evidence of attenuation of SoC effects with age suggests that, to the extent that interindividual variation is driven by periconceptional environmental factors, screens for putative MEs (including ESS and SIV) in adult tissues used as a reference in this analysis may be missing metastable regions which are more pronounced earlier in the life course. Evidence of much larger SoC effect sizes at known MEs in both Gambian cohorts supports this. Furthermore, identification of putative human SIV-MEs requires analysis of datasets with tissues derived from different germ layers for each individual. Such datasets are expensive to acquire and rare, meaning that the few published ME screens are likely to be underpowered due to small sample sizes.

The observed attenuation of SoC effects with age has implications for detecting the effect of periconceptional exposures on DNAm in samples collected beyond the neonatal and early childhood periods, an important consideration for epigenetic epidemiological studies since non-persisting methylation differences could still have a significant impact on early developmental trajectories with lifelong consequences (*Vukic et al., 2019*; *Simpkin et al., 2015*). The methylome changes with age, a phenomenon sometimes referred to as 'epigenetic drift' (*Teschendorff et al., 2013*). Attenuation of SoC effects could be a consequence of localised or global epigenetic drift or other processes such as clonal selection (*Teschendorff and Relton, 2018*).

Interestingly, more SoC-CpGs replicated in the smaller ENID (5–7 yr) dataset than in the older EMPHASIS (7–9 yr) cohort (n = 157 vs. 61, respectively). This could be due to age or other cohort-specific differences, or it might reflect differences in seasonality such as secular changes in the intensity and duration of the rainy season.

Intra-individual methylation states at SoC-associated loci are highly correlated across loci despite being distributed throughout the genome, suggesting that a common mechanism is at play. MEs located within intracisternal A particle (IAP) TEs have been the focus of many studies of SIV in mice, the *Agouti* viable yellow locus being a paradigm example. While one study found that methylation does not covary across different murine IAP MEs in the absence of environmental perturbations (*Kazachenka et al., 2018*), another recent study identified several IAP MEs located on different chromosomes which are modified by the same cluster of KRAB zinc finger proteins (KZFP) (*Bertozzi et al., 2020*), demonstrating that genetic mechanisms for the simultaneous epigenetic regulation of multiple loci do exist.

Potential insights into mechanisms linking periconceptional environment to DNAm changes in postnatal tissues come from our investigations of the methylation status and genomic context of SoC-CpGs.

First, we observed strong enrichment for gDMRs, with 14% of SoC-CpGs overlapping 19 gDMRs hypomethylated in sperm and hypermethylated in oocytes. A minority of these show evidence of PofOm persisting in postnatal tissues. This observation aligns with a growing body of evidence linking early environment, notably nutritional factors involved in one-carbon (C1) metabolism, with methylation at imprinted regions (*Monk et al., 2019*; *James et al., 2018b*). Indeed we have previously noted an association between SoC and several C1 metabolites at a maternally imprinted region at the small non-coding RNA *VTRNA2-1* (*Silver et al., 2015*), consistent with evidence of 'polymorphic imprinting' linked to prenatal environment at this locus (*Zink et al., 2018*; *Carpenter et al., 2018*). Furthermore, we previously found strong enrichment for proximal binding sites of several TFs associated with the maintenance of PofOm in the early embryo at MEs detected in a WGBS screen (*Kessler et al., 2018*). We were only able to replicate enrichment for one of these, ZFP57, at SoC-CpGs in this study. This may reflect the relatively small proportion of PofOm loci in the set of SoC-CpGs, or factors related to the limited methylome coverage of Illumina arrays. More targeted experimental work is required to determine the extent of SoC effects at imprinted loci, especially given our observation that SoC-CpGs are often proximal to ERV TEs that have recently been shown to drive the establishment of germline-derived maternal PofOm (*Bogutz et al., 2019*). Hotspots with evidence of PofOm could be driven by an environmentally sensitive gain of methylation on the paternal allele that is propagated through development, incomplete reprogramming on the maternal allele leaving residual traces of methylated cytosines, or modest de novo methylation at some later point. A deeper understanding of mechanisms will require further investigation in cell and animal models.

Second, our observation of enrichment for proximity to ERV1 TEs at SoC-CpGs aligns with our previous finding at MEs (*Silver et al., 2015*; *Kessler et al., 2018*), and is notable since most environment-sensitive mouse MEs are associated with IAPs (which are rodent-specific ERVs) (*Kazachenka et al., 2018*). KZFP-mediated repression of TEs including ERVs has also been proposed as a driver of the rapid evolution of gene regulation (*Cavalli and Heard, 2019*). The KZFP ZFP57 is particularly interesting in this respect since its binding to DNA is linked both to repression of TEs and to the maintenance of genomic imprints in the pre-implantation embryo (*Imbeault et al., 2017*; *Shi et al., 2019*). We previously identified a putative SoC-associated DMR in the *ZFP57* promoter in blood from younger Gambian infants (mean age 3.6 months) (*Silver et al., 2015*). It is possible that non-replication of the SoC association at *ZFP57* in the older Gambian cohort (and of the *VTRNA2-1* SoC-DMR mentioned above which was also identified in younger infants) reflects the more general attenuation of SoC effects described above. Interestingly there is some evidence that the *ZFP57* DMR, which lies 3 kb upstream of the transcription start site, is established in the early embryo (*Van Baak et al., 2018*; *Monteagudo-Sánchez et al., 2020*), and that DNAm at this locus measured in neonatal blood is associated with maternal folate levels (*Gonseth et al., 2015*; *Amarasekera et al., 2014*; *Irwin et al., 2019*). Given the important function of ZFP57 in pre-implantation methylation dynamics, its potential role as an environmentally sensitive regulator of multi-locus DNAm effects remains an open question.

Third, DNAm at SoC-CpGs is enriched for intermediate methylation states. Intermediate methylation has also been observed at MEs in Gambians and in non-Africans (*Finer et al., 2016*; *Waterland et al., 2010*; *Dominguez-Salas et al., 2014*; *Kühnen et al., 2016*), and this coincides with a similar observation at MEs in post-gastrulation embryonic tissues (*Kessler et al., 2018*). Intermediate methylation appears to be predominantly driven by variegated (intercellular) differences in methylation status

within a sampled tissue, rather than by allele-specific methylation (*Elliott et al., 2015*). Observed differences in aggregate methylation at SoC-CpGs could thus reflect a direct influence of periconceptional environment on the establishment and maintenance of DNAm states in the early embryo at the cellular level, or 'epigenetic selection' whereby epigenetically distinct cells form a substrate for clonal selection during development, for example, as a potential adaptation to differential metabolic exposures (*Tobi et al., 2018*). Stronger enrichment for putative MEs exhibiting SIV compared to those identified through ESS supports establishment in the post-implantation embryo, since methylation differences at ESS loci are presumed to originate in the pre-implantation embryo prior to embryo cleavage in MZ twins (*Van Baak et al., 2018*).

The largest SoC-CpG cluster with 7 CpGs is on chromosome 15 and falls within the second intron of *IGF1R*. This bears the hallmarks of being a promoter or active/poised enhancer in multiple cell lines (*Appendix 1—figure 14*). Six CpGs in the *IGF1R* cluster have evidence of PofOm and show a relatively large SoC effect size (median SoC methylation amplitude) of 6.1%. *Zink et al., 2018*, were unable to demonstrate PofO allele-specific expression in this region although others have found evidence of maternal imprinting of an intronic lncRNA at this gene in cancerous cells (*Kang et al., 2015*; *Sun et al., 2014*). Loss of IGF1 receptors gives rise to a major decrease in expression at multiple imprinted genes in mice, suggesting a pathway by which *IGF1R* might regulate growth and metabolism during early development (*Boucher et al., 2014*). IGF1R signalling is implicated in fetal growth, glucose metabolism and cancer (*Randhawa and Cohen, 2005*; *Aguirre et al., 2016*; *Larsson et al., 2005*), and DNAm differences at *IGF1R* have been observed in birthweight-discordant adult twins (*Tsai et al., 2015*). Another SoC-CpG with evidence of PofOm, also on chromosome 15, falls within the known Prader-Willi syndrome-associated paternally expressed imprinted region.

Our observation that 109 out of 259 SoC-CpGs have been associated with sex in previous studies is intriguing. Given the relatively small number (compared to array size) of autosomal sex-linked loci identified in large studies on the Illumina 450k array (*Singmann et al., 2015*; *Shah et al., 2014*), this represents a very strong enrichment. We replicated a significant sex association at these loci in the ENID cohort analysed in this study. Our regression analyses were adjusted for sex, and additional sensitivity analyses with DNAm pre-adjusted for sex strongly suggest that our main findings are not confounded by sex. Interestingly, sex-associated loci are enriched at imprinted regions and sex discordance at autosomal CpGs has been linked to androgen exposures in utero (*Singmann et al., 2015*; *Suderman et al., 2017*). There is also evidence of sex differences in methylation at *DNMT3A/B* and *TET1* genes involved in de novo methylation and de-methylation pathways (*Singmann et al., 2015*; *Suderman et al., 2017*), suggesting a possible interaction between sex-linked epigenetic changes and periconceptional environment during reprogramming in the early embryo. A deeper understanding of potential mechanisms underpinning the observed enrichment of sex effects at loci associated with periconceptional environment requires further functional analysis in cell and/or animal models.

DNAm is influenced by genotype and the latter is therefore a potential confounder when studying the effects of environmental exposures in human populations. A strength of our quasi-randomised Gambian seasonal model is that it minimises the potential for genetic confounding of modelled seasonal DNAm patterns, on the assumption that the timing of conceptions is not linked to genetic variants influencing DNAm. However, it is still possible that such variants might confound our observations, for example, if they promote embryo survival under conditions of environmental stress. We tested this possibility using genetic data available for the EMPHASIS cohort and found no evidence of SoC-associated genetic variants driving interindividual methylation differences at SoC-associated loci in *cis* or *trans*.

However, we did find that half of SoC-CpGs had at least one associated mQTL, indicating the presence of independent additive effects of environment and genetics at these loci, as has been suggested previously at other loci sensitive to pre/periconceptional nutritional exposures (*Tobi et al., 2012*; *Saffari et al., 2020*). We have previously argued that the definition of MEs should be extended to include genomic regions whose DNAm state is under partial but non-deterministic genetic influence in genetically heterogeneous human populations (*Kessler et al., 2018*), and we would argue that the above observations at SoC-CpGs that exhibit many of the characteristics of MEs support this. Further analysis in larger datasets with genome sequencing data combined with functional analysis using cell models will be required to fully understand the relative contributions of environment and genetics to DNAm variation at regions of the type highlighted in this study.

Further work is required to investigate the functional relevance of DNAm changes at SoC-CpGs, some of which are relatively small (around 4% SoC amplitude). However, we can speculate that observed methylation changes, which may reflect changes in the chromatin landscape, have the potential to influence gene expression and early development, since our chromatin state analysis predicts that many overlap regions with functional significance in H1 ESCs and fetal tissues. A similarly modest DNAm change at a locus in the *POMC* gene that is associated with SoC in blood has been linked to obesity risk in German children and adults (*Kühnen et al., 2016*; *Kuehnen et al., 2012*), and to differential TF binding and differences in *POMC* expression (*Kuehnen et al., 2012*). Furthermore, blood DNAm at a putative ME within the *PAX8* gene has been linked to SoC in Gambian infants (*Dominguez-Salas et al., 2014*) and is associated with thyroid volume and function in Gambian children, and with certain maternal nutrition biomarkers at conception (*Candler et al., 2021*). *PAX8* DNAm is also associated with expression of the anti-sense gene *PAX8-AS1* (alternatively known as *LOC654433*) in thyroid tissue (*Candler et al., 2021*). No loci from either study overlap array background in this study. More generally, DNAm at SIV loci associated with periconceptional environment has been associated with a number of diseases including Alzheimer's, cancer, rheumatoid arthritis, and schizophrenia (*Gunasekara and Waterland, 2019*).

We have previously shown that several metabolites involved in C1 methylation pathways show significant seasonal variation in maternal blood plasma in this population which is largely dependent on subsistence farming (*Dominguez-Salas et al., 2013*; *Dominguez-Salas et al., 2014*). While we suspect that seasonal differences in nutrition are likely drivers of the effects that we have observed, analysis of the links between maternal nutritional biomarkers and DNAm is challenging due to the complex interdependence of C1 metabolites acting as substrates and cofactors within C1 pathways (*James et al., 2018a*). Furthermore, it is possible that DNAm differences are linked to other aspects of Gambian seasonality, such as variation in pesticide use or infectious disease burden (*Moore et al., 1999*; *Hernandez-Vargas et al., 2015*).

Measured DNAm values at SoC-CpGs and matched controls have previously been reported to be reliable, in strong contrast to the majority of probes overlapping the 450k and EPIC arrays that have been found to have low test-retest reliability (*Sugden et al., 2020*). This likely reflects increased interindividual variability and/or intermediate DNAm at SoC-CpGs and controls (*Logue et al., 2017*; *Xu and Taylor, 2021*).

Limitations of this analysis include a lack of genetic data for the ENID cohort, so that our genetic analyses are restricted to the older EMPHASIS cohort where SoC effects are smaller. Furthermore, comparator data on MEs and gamete, embryo and PofO methylation are derived from non-African individuals so we cannot assess the influence of ethnicity on these analyses. We also note that there is some inter-relatedness in this study population which practices polygamy. However, of the 199 individuals with available data on paternal identity in the ENID cohort, 2% have a shared father suggesting that this is unlikely to influence the main findings.

There is increasing interest in the phenomenon of methylation variability as a marker of disease and of prenatal adversity, and in genetic variation as a potential driver of methylation variance (*Ek et al., 2018*). This raises the possibility that certain genetic variants could have been selected through their ability to enable graded, environmentally responsive methylation patterns at MEs and SoC-associated loci that are able to sense the periconceptional environment, record the information, and adapt the phenotype accordingly. Our gene-environment interaction analysis likely lacked power to detect gene-environment (SoC) interactions, so that it was not possible to investigate the relative contributions of stochastic, environmental, and genetically mediated variance effects on the establishment of periconceptional SoC-associated methylation states in this study. Nonetheless, the proposition that environmentally sensitive epigenetic signals are selected through their ability to direct phenotypic development to better fit the anticipated future environment, leading to maladaptation and future disease if the environment changes, is intriguing and worthy of further investigation (*Fleming et al., 2018*).

## Materials and methods

### Gambian cohorts and sample processing

Detailed descriptions of the Gambian cohorts analysed in this study are published elsewhere (*Moore et al., 2012*; *Chandak et al., 2017*). Briefly, for the ENID 2 yr dataset, blood samples from 233 children aged 24 months (median [inter-quartile range, IQR]: 731 [729,733] days of age) were collected from participants in the **E**arly **N**utrition and **I**mmune **D**evelopment ('ENID') trial (*Moore et al., 2012*), born in 2011 and 2012. DNA was extracted, bisulfite-converted, and hybridised to Illumina Human-Methylation450 (hereafter 'HM450') arrays following standard protocols (see *Van Baak et al., 2018*, for further details). *N* = 138 ENID participants with 2 yr HM450 data who were enrolled in a follow-up study with DNA collected at 5–7 yr (6.2 yr [5.7,6.6]) had DNA bisulfite-converted and hybridised to Illumina Infinium Methylation EPIC (hereafter 'EPIC') arrays. For the EMPHASIS cohort, DNA was extracted from blood samples from 289 Gambian children aged 7–9 yr (9.0 [8.6,9.2] years) participating in the **E**pigenetic **M**echanisms linking **P**re-conceptional nutrition and **H**ealth **As**sessed in India and **S**ub-Saharan Africa ('EMPHASIS') study (*Chandak et al., 2017*), born between 2006 and 2008. DNA was bisulfite-converted and hybridised to EPIC arrays, again using standard protocols.

For the ENID cohort, date of conception was calculated from fetal gestational age estimates obtained by ultrasound at the mother's first 'booking' appointment. The same method was used for the EMPHASIS cohort, except for *n* = 71 pregnancies that were >24 weeks' gestation at booking meaning that GA could not be accurately determined by ultrasound (*Chandak et al., 2017*; *Owens et al., 2015*). In this case date of conception was calculated as date of birth minus 280 days which is the average gestational length for this population.

Samples from the ENID 2 yr dataset were processed in two batches with each batch covering conception dates throughout the year. Samples from the ENID (5–7 yr) and EMHASIS (7–9 yr) datasets were processed in single batches.

### Methylation array pre-processing and normalisation

Raw intensity IDAT files from the HM450 and EPIC arrays were processed using the *meffil* (*Min et al., 2018*) package in R (v3.6.1) using standard *meffil* defaults. Briefly, this comprised probe and sample quality control steps (filtering on bisulfite conversion efficiency, low probe detection p-values and bead numbers, high number of failed samples per probe, high number of failed probes per sample); methylation-derived sex checks; removal of ambiguously mapping (i.e. cross-hybridising) probes (*Chen et al., 2013*); removal of probes containing SNPs at the CpG site or at a single base extension; and removal of non-autosomal CpGs. Following filtering, methylation data was normalised with dye-bias and background correction using the *noob* method, followed by *Functional Normalisation* to reduce technical variation based on PCA of control probes on the arrays (*Fortin et al., 2014*). After pre-processing and normalisation, methylation data comprised methylation beta values for 421,026 CpGs on the HM450 array for the 233 individuals from the ENID 2 yr cohort, and 802,283 CpGs on the EPIC array for 289 individuals from the EMPHASIS cohort; 391,814 CpGs intersecting both arrays were carried forward for statistical analysis of these two datasets. Finally, SoC-CpGs, matched and random controls, all of which were present in the ENID (5–7 yr) EPIC dataset, were included in the longitudinal analysis.

### Statistical modelling

Variation of DNAm with date of conception was modelled using Fourier regression (*Rayco-Solon et al., 2005*). This models the relationship between a response variable (here DNAm) and a cyclical predictor (date of conception). The effect of the latter is assumed to be cyclical due to annually varying seasonality patterns, so that the modelled effect for an individual conceived on the 31 December should be 'close' to that for an individual conceived on the 1 January. This is achieved by deconvolving the conception date (predictor) into a series of pairs of sin and cosine terms, and obtaining estimates for the regression coefficients $\beta$ and $\gamma$ in the following model:

$$M_{ij} = \alpha_{0j} + \sum_{k=1}^{m} \alpha_{ik} + \sum_{r=1}^{n} \left[ \beta_{rj} \sin\left(r\theta_i\right) + \gamma_{rj} \cos\left(r\theta_i\right) \right] + \varepsilon_{ij}$$

where, for individual i and CpG *j*:

$M_{ij}$ is the logit-transformed methylation beta value (**Du et al., 2010**);
$\alpha_{0j}$ is an intercept term;
$\alpha_{ik}$ is the $k$th of $m$ adjustment covariates;
$\theta_i$ is the date of conception in radians in the interval $[0, 2\pi]$, with 1 January = 0 and 31 December = $2\pi$, modelled as $n$ pairs of Fourier terms, $\sin\theta_i\cos\theta_i + \ldots + \sin n\theta_i\cos n\theta_i$;
$\beta$rj and $\gamma$rj are the estimated regression coefficients for the $r$th sin and cosine terms, respectively;
and $\varepsilon_{ij}$ is the error term.

With a single pair of Fourier terms ($n = 1$), this gives a sinusoidal pattern of variation, with a single maximum and minimum whose phase (position in the year) and amplitude (distance between methylation maximum and minimum) is determined by $\beta_1$ and $\gamma_1$, with the constraint that the maximum and minimum are 6 months apart. More complex patterns of seasonal variation are afforded by higher frequency pairs of Fourier terms ($r > 1$).

For this analysis we modelled the effect of date of conception using a single pair of Fourier terms ($n = 1$) and assessed goodness-of-fit by comparing full and covariate-only models using likelihood ratio tests (LRTs). For all datasets, covariates included child sex, and the first six PCs obtained from unsupervised PCA of the normalised methylation $M$-values. The PCs were used to account for unmeasured and measured technical variation (due to bisulfite conversion sample plate, array slide, etc.) and cell composition effects (see **Supplementary file 1p-r**). Additional checks confirmed no seasonal variation in estimated white cell composition in the ENID 2 yr and EMPHASIS 7–9 yr cohorts used in the main analysis (see below); 450k Sentrix Column was included as an additional adjustment covariate for the ENID 2 yr cohort since this was not robustly captured by any of the first 6 PCs (**Supplementary file 1p**). Child age was included as an additional adjustment covariate for the ENID 5–7 yr and EMPHASIS cohorts, as was maternal nutritional intervention group (see **Chandak et al., 2017**; **Saffari et al., 2020** for further details).

For each CpG $j$, coefficient estimates $\beta_j$, $\gamma_j$ were determined by fitting regression models using *lm*() in R. Model goodness-of-fit was determined by LRT using *lrtest*() in R, comparing the full model including Fourier terms, with a baseline covariates-only model. A model p-value, $p_j$ was then derived from the corresponding LRT chi-squared statistic. Thus for a given threshold, $\alpha$, $p_j<\alpha$ supports rejection of the null hypothesis that for CpG $j$, the full model including the effect of seasonality modelled by one pair of Fourier terms fits no better than the covariate-only model at the $\alpha$ level.

To reduce the influence of methylation outliers, methylation values greater than three times the methylation IQR beyond the 25th or 75th percentiles were excluded prior to fitting the models. This resulted in 382,095 (267,369) outliers, or an average of 0.98 (0.68) outliers per CpG being removed from the ENID and EMPHASIS methylation datasets, respectively.

Note that our observation of attenuation of SoC effect sizes with age at SoC-CpGs when comparing DNAm data at aged 2 yr (ENID) and 7–9 yr (EMPHASIS), supported by analysis of longitudinal data (ENID 5–7 yr), suggested that a combined meta-analysis of ENID and EMPHASIS datasets would have reduced power to detect SoC effects at 2 years of age.

## Sensitivity analyses

We performed sensitivity analyses following the same Fourier regression modelling strategy as outlined above, but with (i) methylation estimated cell counts using *estimateCellCounts*() from *minfi* (v1.30.0); (ii) known batch and technical covariates; and (iii) village ID included as additional covariates. See **Supplementary file 1s** for further details on regression models used in the sensitivity analyses. For each sensitivity analysis we tested to see if Fourier regression coefficients estimated in the main analysis fell within 95% CIs in the adjusted models.

## Inflation of test statistics

The concept of genomic inflation rests on the assumption that a relatively small number of loci will be associated with the exposure (or disease/outcome) of interest. Test statistics for the ENID (2 yr) cohort did show signs of genomic inflation (lambda = 1.33), suggesting a potential effect of SoC on global methylation levels (**Zheng et al., 2017**). A similar level of inflation was observed before in a study looking at the effect of periconceptional folate on DNAm (**Gonseth et al., 2015**). There is also evidence of global and/or multi-locus effects of folate in other studies, including an RCT of folate supplementation in pregnancy (**Irwin et al., 2019**), and there are many other examples including

studies investigating the effect of mutations in the *MTHFR* gene (see review by *Crider et al., 2012*). While we do not know if the SoC associations we observe in our cohorts are driven by seasonal differences in folate, we do observe significant seasonal differences in multiple C1 metabolites including folate in our population (*Dominguez-Salas et al., 2013*; *Dominguez-Salas et al., 2014*), so that SoC may serve as a proxy for multi-locus C1 metabolite effects in our analyses.

We tested for potential SoC effects on global methylation by analysing methylation differences at LINE1 and Alu elements using REMP (v1.16.0), a recently published method to predict methylation at repetitive elements from Illumina array data (*Zheng et al., 2017*). We used default cutoffs for reliability suggested by the authors. This analysis confirmed a small but significant effect of increased methylation in rainy vs. dry conceptions at LINE1 and Alu elements (mean rainy season increase 0.02% and 0.01%, respectively, both Wilcoxon $p < 2.2 \times 10^{-16}$) suggesting a SoC effect on global methylation levels. This contrasted with a significant but extremely small effect in the opposite direction across array background ($-5 \times 10^{-4}$%; $p < 2.2 \times 10^{-16}$; see *Appendix 1—figure 15*), supporting evidence from previous studies of a potential effect of peri/preconception nutrition-related exposures on global methylation levels.

## Identification of SoC-CpGs

For the ENID 2 yr cohort, p-values, $p_j$, were used to compute a FDR for each CpG accounting for multiple testing (assuming 391,814 independent tests corresponding to the number of loci in array background) using *p.adjust*() in R with method = 'fdr'. Following the rationale described in the main text, SoC-associated CpGs with a SoC amplitude <4% in the ENID cohort were then excluded to form the final set of 259 'SoC-CpGs' (see *Table 2*).

## Selection of control CpGs

SoC-CpGs are enriched for intermediate methylation states (*Figure 4B*), so that there is a risk that some downstream analyses reflect the distributional properties of these loci, rather than factors associated with their putative establishment at periconception. For this reason we identified a set of 'matched control' CpGs that were selected to have similar methylation beta distributions to SoC-CpGs (and additional SoC-associated CpGs with amplitude <4%) in the ENID 2 yr dataset. Matched controls were drawn from array background (excluding SoC-associated CpGs and known MEs/ESS/SIV CpGs), with one matched control identified for each of the 768 SoC-associated CpGs. Alignment of control and SoC-CpG methylation distributions was achieved using a two-sided Kolmogorov-Smirnov test for divergence of cumulative distribution functions (*ks.test()* in R) with a p-value threshold $p > 0.1$. Examples are given in *Appendix 1—figure 16*, along with a comparison of sample mean distributions.

An additional 768 **random control** CpGs were randomly sampled from array background, again excluding SoC-CpGs and known MEs/ESS/SIV CpGs.

## CpG sets considered in analyses

Summary information on external datasets considered in the analyses is provided in *Table 3*. Further information on these is provided below.

1. *1881 putative ME* CpGs overlapping array background from one or more of the following curated sets of loci: putative MEs exhibiting SIV identified in a multi-tissue WGBS screen in individuals of European and African-American decent described in *Kessler et al., 2018*; and CpGs exhibiting 'ESS' and/or SIV derived from analysis of 450k data from individuals of European decent, as described in *Van Baak et al., 2018*.
2. *699 PofOm* CpGs overlapping array background from 229 regions with PofOm identified in Supplementary Table S1 from *Zink et al., 2018*. PofOm identified in peripheral blood from Icelandic individuals.
3. *RRBS early stage embryo data* from Chinese embryos described in *Guo et al., 2014*, downloaded from GEO (accession number GSE49828). Only CpGs covered at ≥10× in pre-gastrulation ICM and/or post-gastrulation embryonic liver were considered in this analysis. Further details are provided in *Kessler et al., 2018*.
4. *Sperm methylation data* from Japanese donors described in *Okae et al., 2014*, downloaded from the Japanese Genotype-Phenotype Archive (accession number S00000000006). Only CpGs covered at ≥10× were considered in this analysis.

5. gDMRs, defined as contiguous 25 CpG regions that were hypomethylated (DNAm mean + 1 SD < 25%) in one gamete and hypermethylated (DNAm mean − 1 SD >75%) in the other, were previously identified from WGBS data by *Sanchez-Delgado et al., 2016*. Persistence of PofOm to the blastocyst and placental stages was established by identifying overlapping intermediately methylated regions in the relevant embryonic tissues, with confirmation of PofOm expression at multiple DMRs (*Sanchez-Delgado et al., 2016*). Japanese and US donors. See *Sanchez-Delgado et al., 2016*, for further details.

6. *TEs (ERVs)* determined by RepeatMasker were downloaded from the UCSC hg19 annotations repository.

7. ZFP57, TRIM28, and CTCF TF binding sites identified from ChIP-seq in human embryonic kidney and human embryonic stem cells used in this analysis are described in *Kessler et al., 2018*.

## Cluster-based adjustments

Many SoC-CpGs cluster together and this could influence some analyses. For example, methylation at CpGs may be highly correlated, which could influence comparisons of inter-CpG correlations between SoC-CpGs and controls (*Figure 3D*). Also enrichment tests are likely to be influenced by neighbouring CpGs that together constitute a single 'enrichment signal' proximal to a particular genomic feature (e.g. a TF binding site).

To account for this, cluster-adjusted analyses used 'de-clustered' CpG sets that were constructed as follows:

1. Create CpG clusters formed from adjacent CpGs where each CpG is within 5 kbp of the nearest neighbouring CpG (see *Appendix 1—figure 5* for a justification of this threshold).
2. Construct de-clustered test set by randomly sampling a single CpG from each cluster; non-clustered 'singleton' CpGs are always selected.

In the case of SoC-CpGs, the set of 259 non-clustered CpGs were reduced to 161 CpGs after de-clustering.

## Chromatin state and histone (H3K) mark analysis

Data on chromatin states predicted by the ChromHMM 15-state model (*Ernst and Kellis, 2012*) in H1 ESCs, and three fetal tissues (brain, muscle, and small intestine) derived from three different germ layers, generated by the *Roadmap Epigenomics Consortium et al., 2015*, was downloaded from the Washington University Roadmap Epigenomics repository; 15-state model predictions were collapsed to eight states for visualisation purposes, with sub-classifications described in the *Figure 5* caption.

Overlaps with H3K marks for each of the above tissues were assessed using the *annotatr (v1.10.0)* package in R to interrogate the same Roadmap Epigenomics ChIP-seq data used by ChromHMM for chromatin state prediction.

## Additional modelling of seasonal variation in blood cell composition

Cell count estimates using the Houseman method (*Jaffe and Irizarry, 2014*) were obtained using the *estimateCellCounts*() from *minfi* (v1.30.0) in R. Seasonal variation in blood cell composition was then modelled by Fourier regression with one pair of Fourier terms and sex (ENID + EMPHASIS) and age (EMPHASIS only) as adjustment covariates. Fitted models indicated no marked seasonal differences within and between cohorts (*Appendix 1—figure 17*).

## Genetic association analyses

### Genotype data

mQTL and related SoC association analyses were performed on all 284 individuals from the EMPHASIS (7–9 yr) cohort for which we had QC'd genotype data; 259 SoC-CpGs plus sets of 259 matched and random control CpGs were considered in this analysis. Subjects were genotyped using the Illumina Infinium Global Screening Array-24 v1.0 Beadchip (Illumina, San Diego, CA) following standard protocols. Array-derived genotypes were pre-phased using SHAPEITv2 and imputation was performed using IMPUTEv2.3.2 on 1000 genomes phase 3 data. Further details are provided in *Saffari et al., 2020*. SNPs with a MAF ≤10% were excluded, along with those with an IMPUTE 'info' metric ≤0.9, a stringent threshold to ensure maximum confidence in imputation quality. Imputed SNPs were then pruned (using plink v1.90 -indep-pairwise with window size 50, step size five and $r^2$ threshold of 0.8)

to remove SNPs in strong LD. Finally, to minimise the influence of low-frequency homozygous variants in linear models, analysis was restricted to SNPs with 10 or more homozygous variants, resulting in a final dataset comprising 2,609,310 SNPs.

## Identification of mQTL and 'GxE' SNPs

mQTL analysis was performed using the *GEM* package (v1.10.0) from R Bioconductor (***Pan et al., 2016***).

SNP effects on methylation were modelled as follows:

$$M_j \sim covs + G$$

where $M_j$ is the methylation *M*-value for CpG *j*, *G* is the SNP genotype coded as allelic dosage (0,1,2) and covs correspond to the adjustment covariates used in the main EMPHASIS analysis (PCs 1–6, child age, sex, and intervention status).

GxE (SoC) effects were modelled as follows:

$$M_j \sim covs + \sin\theta + G + G \times \sin\theta$$

when $\sin\theta$ is the most significant Fourier term in the main SoC analysis.

 OR

$$M_j \sim covs + cos\theta + G + G \times \sin\theta$$

when $\cos\theta$ is the most significant Fourier term in the main SoC analysis.

Here, *G* and covs are as described above.

Since most mQTL effects are known to act in *cis* (***Gaunt et al., 2016***), in order to maximise power, a two-step procedure was used to identify significant mQTL:

(i) *Identification of cis-mQTL* passing FDR < 5% considering the reduced set of SNPs within 1 Mbp of a CpG in the set to be analysed (SoC-CpGs, matched or random controls); (ii) *identification of trans-mQTL* passing FDR < 5%, considering the full set of 2.6 M SNPs.

## Calculation of methylation variance explained by mQTL

For each CpG *j*, total methylation variance explained was calculated for the following model:

$$M_j \sim covs + mQTL_{j1} + mQTL_{j2} + ...mQTL_{j_n}$$

where covs are as defined above, and $mQTL_{jn}$ is the genotype (coded 0, 1, 2) of the *n*th mQTL mapping to CpG *j*. Methylation variance explained was calculated from the model adjusted $R^2$ value, $adjR^2_{mQTL}$, to account for different levels of model complexity due to the differing number of mQTL identified for each CpG.

A final estimate of mQTL methylation variance explained was obtained by subtracting variance explained by the covariate-only model:

$$\Delta adjR^2 = adjR^2_{mQTL} - adjR^2_{cov}$$

where $adjR^2_{cov}$ is the adjusted $R^2$ for the covariate-only model.

$$M_j \sim covs.$$

## SoC association analysis

Potential confounding of SoC-DNAm signals by SoC-associated genetic variants was assessed by analysing SoC associations with all 2771 SoC-CpG-associated mQTL identified in the mQTL analysis (see *Table 4*). SoC association analysis was performed under an allelic model using -assoc with *plink* v1.90. Significant mQTL-SoC associations were identified using FDR < 5%, which assumes independence of all mQTL-SoC associations. We also considered a more liberal multiple testing correction threshold: $p_{Bonf} = 0.05/130$, assuming complete dependence of all *cis*-mQTL mapping to each of the 130 SoC-CpG with an associated mQTL.

## Genetic ancestry sensitivity analysis

An investigation of population structure in the EMPHASIS cohort was conducted by first performing a PCA using -pca in plink v1.90. The PCA was performed on non-imputed data, with LD pruning using -indep-pairwise 50 5 0.2 in plink v1.90, corresponding to an $r^2$ threshold of 0.2 *Weale, 2010*. Evidence for population structure was then obtained by plotting the first four PCs (*Appendix 1—figure 12*). Confirmation of a likely link to Gambian ethnic ancestry in this largely ethnically Mandinka region of Gambia followed our observation that a distinct cluster (*Appendix 1—figure 12*) was primarily made up of individuals from a single, predominantly ethnic Fula village.

A sensitivity analysis to check the effect of accounting for ancestry differences was performed by repeating the main analysis with the following additional covariates.

1. EMPHASIS (7–9 yr) cohort

We adjusted for genetic ancestry directly using the first two genetic PCs identified in the genetic PCA.

2. ENID (2 yr) cohort

Since no genetic data is available for this cohort, and since individuals from this cohort are drawn from the same villages as the EMPHASIS cohort on which we did the genetic PCA, we reasoned that village of origin is a useful proxy for genetic ancestry in our population. We therefore included an additional covariate dichotomised according to whether or not an individual was one of the nine who came from the predominantly Fula village identified as genetic outlier in the EMPHASIS PCA analysis.

## Sensitivity analysis to investigate confounding by infant sex

Our finding that multiple SoC-CpGs were associated with infant sex in previous EWAS prompted us to perform a sensitivity analysis checking for the possibility of a residual confounding effect due to sex. To do this we regressed out the effect of infant sex at each CpG in the $M$-value methylation matrix, prior to the main regression analysis. We then re-ran the Fourier regression analysis with and without an additional adjustment for infant sex in Fourier regression models. As expected, given that we adjusted for infant sex in the main analysis, this produced near-identical results, suggesting that the main analysis was not confounded by infant sex.

## Bootstrapped confidence intervals

All bootstrapped confidence intervals presented in this paper use 1000 bootstrap samples.

## Acknowledgements

The Gambian ENID trial was jointly funded by the UK Medical Research Council (MRC) and the Department for International Development (DFID) under the MRC/DFID Concordat agreement (MRC Program MC-A760-5QX00). Methylation analysis of ENID samples was supported by the Bill & Melinda Gates Foundation (grant no. OPP1 066947). The Gambian EMPHASIS study is jointly funded by MRC, DFID, and the Department of Biotechnology, Ministry of Science and Technology, India, under the Newton Fund initiative (MRC grant no. MR/N006208/1 and DBT grant no. BT/IN/DBT-MRC/DFID/24/GRC/2015–16). Further support for this analysis was provided by MRC Grant MR/M01424X/1. We acknowledge the work of the full EMPHASIS Study Group (https://www.emphasis-study.org/) in acquiring this data.

## Additional information

### Funding

| Funder | Grant reference number | Author |
| --- | --- | --- |
| Medical Research Council | MC-A760-5QX00 | Matt J Silver<br>Andrew M Prentice |

| Funder | Grant reference number | Author |
|---|---|---|
| Bill and Melinda Gates Foundation | OPP1 066947 | Sophie E Moore Michael N Routledge Zdenko Herceg |
| Medical Research Council | MR/N006208/1 | Matt J Silver Caroline HD Fall Andrew M Prentice |
| Department of Biotechnology, Ministry of Science and Technology, India | BT/IN/DBT-MRC/DFID/24/ GRC/2015-16 | Gririraj R Chandak |
| Medical Research Council | MR/M01424X/1 | Matt J Silver Ayden Saffari Noah J Kessler Andrew M Prentice |

The funders had no role in study design, data collection and interpretation, or the decision to submit the work for publication.

### Author contributions

Matt J Silver, Conceptualization, Formal analysis, Funding acquisition, Investigation, Methodology, Project administration, Software, Supervision, Visualization, Writing - original draft, Writing – review and editing; Ayden Saffari, Formal analysis, Methodology, Software, Writing – review and editing; Noah J Kessler, Formal analysis, Methodology, Software, Visualization, Writing – review and editing; Gririraj R Chandak, Funding acquisition, Project administration, Supervision, Writing – review and editing; Caroline HD Fall, Funding acquisition, Project administration, Writing – review and editing; Prachand Issarapu, Akshay Dedaniya, Investigation, Writing – review and editing; Modupeh Betts, Michael N Routledge, Cyrille Cuenin, Data curation, Writing – review and editing; Sophie E Moore, Data curation, Funding acquisition, Project administration, Writing – review and editing; Zdenko Herceg, Funding acquisition, Writing – review and editing; Maria Derakhshan, Formal analysis, Methodology, Writing – review and editing; Philip T James, Formal analysis, Writing – review and editing; David Monk, Data curation, Investigation, Writing – review and editing; Andrew M Prentice, Conceptualization, Funding acquisition, Investigation, Writing – review and editing

### Author ORCIDs
Matt J Silver  http://orcid.org/0000-0002-3852-9677

### Ethics

Ethics approval and consent to participateEthics approval for the Gambian ENID and EMPHASIS trials was obtained from the joint Gambia Government/MRC Unit The Gambia's Ethics Committee (ENID: SCC1126v2; EMPHASIS: SCC1441). The ENID study is registered as ISRCTN49285450. The EMPHASIS study is registered as ISRCTN14266771. Signed informed consent for both studies was obtained from parents, and verbal assent was additionally obtained from the older children who participated in the EMPHASIS study.

### Decision letter and Author response
Decision letter  https://doi.org/10.7554/eLife.72031.sa1
Author response  https://doi.org/10.7554/eLife.72031.sa2

## Additional files

### Supplementary files
• Transparent reporting form

• Supplementary file 1. Supplementary tables. (a) ENID SoC-associated CpGs (no amplitude filter applied). (b) Seasonal amplitude tests. (c) Inter-cohort change in SoC amplitude at ENID SoC-associated CpGs. (d) SoC-CpGs (ENID SoC-associated CpGs with SoC methylation amplitude >=4%). (e) SoC-CpG clusters and singletons. (f) SoC-CpG cluster sizes. (g) SoC-CpG enrichment for MEs and overlap with ME sub-classes. (h) SoC-CpG enrichment for gDMRs and parent-of-

origin specific methylation. (i) SoC-CpGs overlapping maternally methylated germline DMRs. (j) SoC-CpGs mQTL and their association with SoC. (k) Candidate genes previously associated with periconception / gestational exposures overlapping array background. (l) Look-up of SoC-CpGs in the EWAS Catalog. (m) EWAS Catalog data grouped by trait. (n) SoC-CpG associations with selected variables measured in the ENID 2yr dataset. (o) Look-up of SoC-CpGs in the GWAS Catalog. (p) ENID (2yr): Association test p-values to detect potential residual confounding. (q) EMPHASIS (7-9yr): Association test p-values to detect potential residual confounding. (r) ENID 5-7yr: Association test p-values to detect potential residual confounding. (s) ENID (2yr): Cell count, batch, village sensitivity analyses.

### Data availability

Illumina 450k methylation array data generated from Gambian 2 year olds from the ENID trial is deposited in GEO (GSE99863). Requests to access and analyse the other Gambian methylation datasets (ENID 5-7yr and EMPHASIS 7-9yr) should be submitted to the corresponding author in the first instance. An application would then need to be made to MRC Unit The Gambia's Scientific Coordinating Committee and the Joint MRC/Gambia Government Ethics Committee. Sources and locations of other publicly available data used in this analysis are described in Methods. Bespoke code used in the analysis is available at https://zenodo.org/record/5801480.

The following previously published dataset was used:

| Author(s) | Year | Dataset title | Dataset URL | Database and Identifier |
|---|---|---|---|---|
| Van Baak T | 2018 | DNA methylation in children from The Gambia | https://www.ncbi.nlm.nih.gov/geo/query/acc.cgi?acc=GSE99863 | NCBI Gene Expression Omnibus, GSE99863 |

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

## Appendix 1

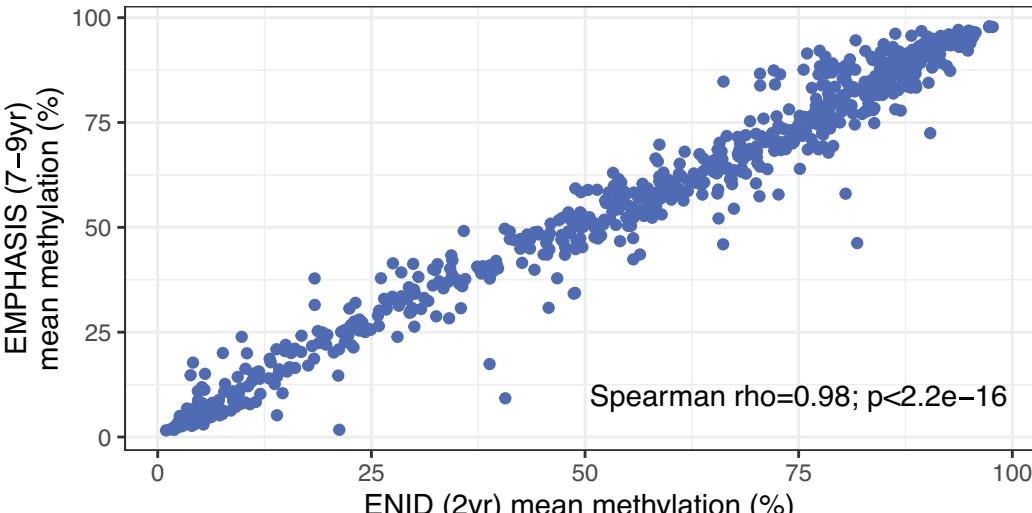

**Appendix 1—figure 1.** Cross-cohort correlation of mean methylation values at 768 season of conception (SoC)-associated loci. Data comprises *n*=233 and *n*=289 individuals from the ENID (2 yr) and EMPHASIS cohorts, respectively. 768 SoC-associated CpGs (false discovery rate [FDR]<5%) identified in the ENID (2 yr) cohort.

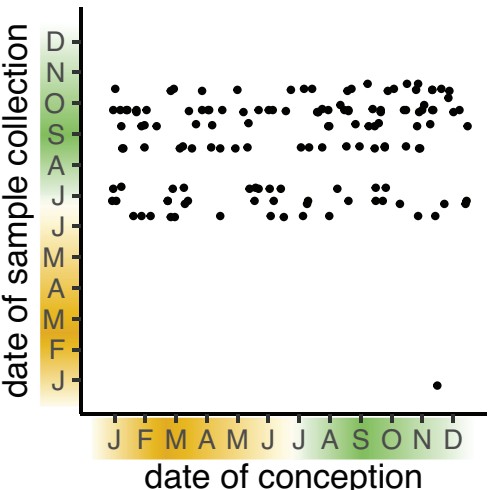

**Appendix 1—figure 2.** Date of conception vs. date of collection for samples analysed in the ENID longitudinal (5–7 yr) dataset. Data comprises samples from *n*=138 ENID participants with methylation data at both 2 yr and 5–7 yr.

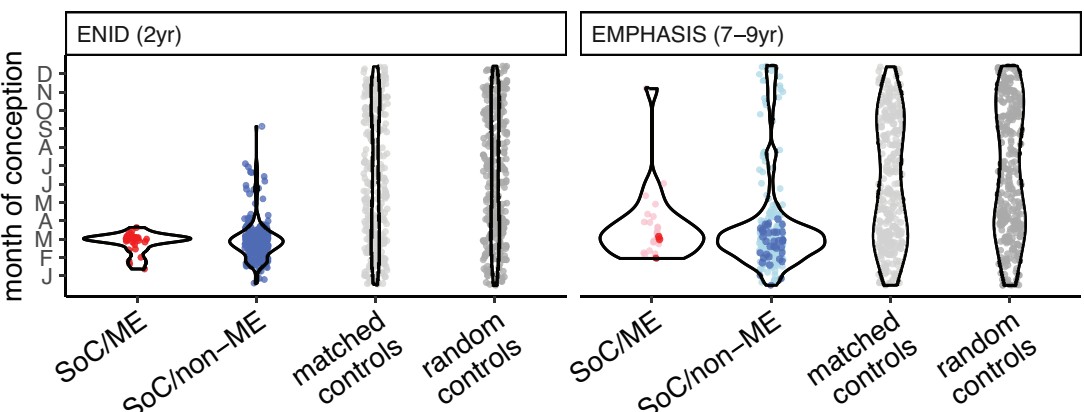

**Appendix 1—figure 3.** Date of conception at modelled methylation minima. Conception dates at methylation minima for loci plotted in *Figure 4A*. Note that since seasonality is modelled by a single pair of Fourier terms, maxima and minima are 6 months apart.

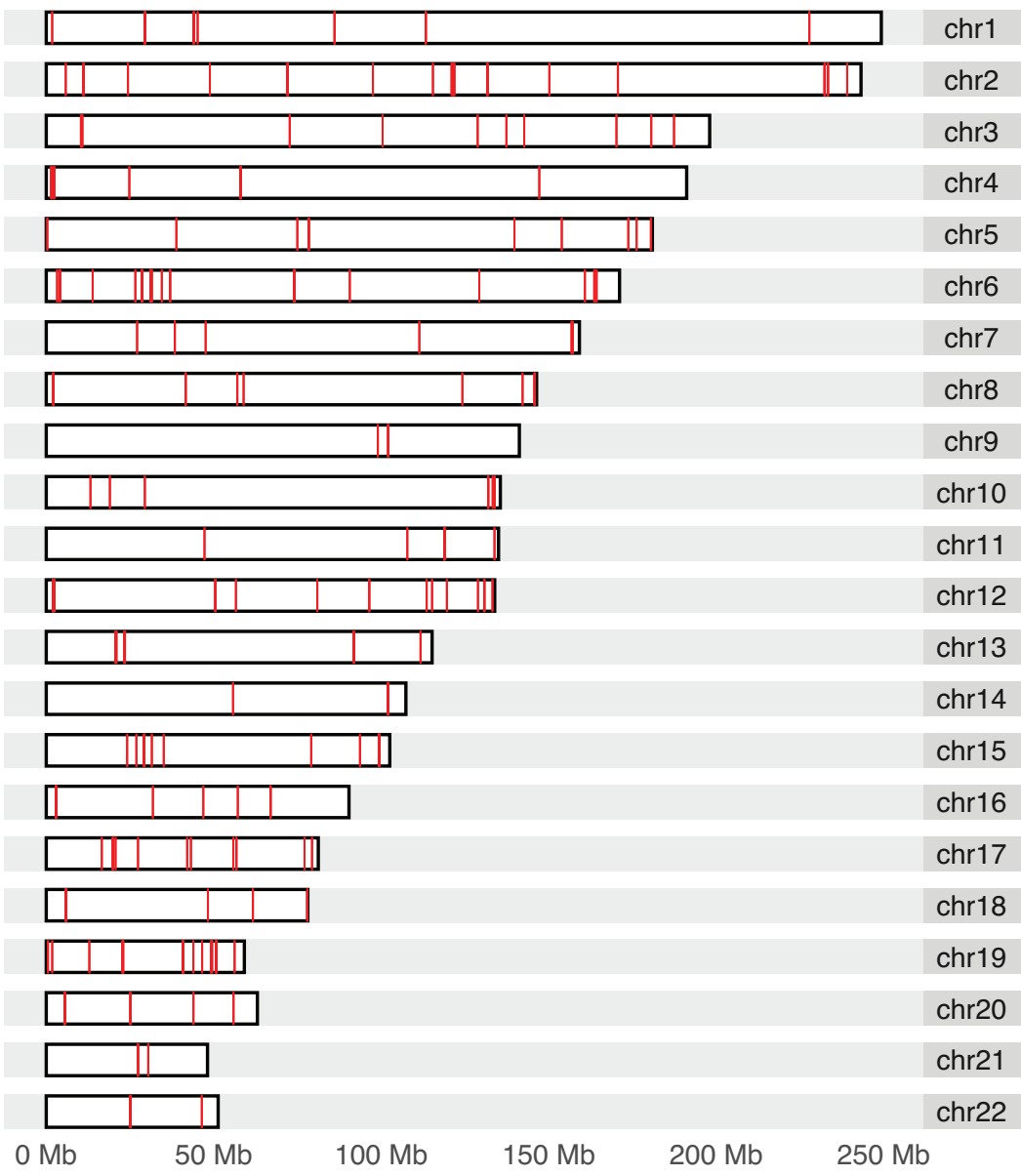

**Appendix 1—figure 4.** Genomic locations (hg19) of 259 SoC-CpGs. Data from *Supplementary file 1D*.

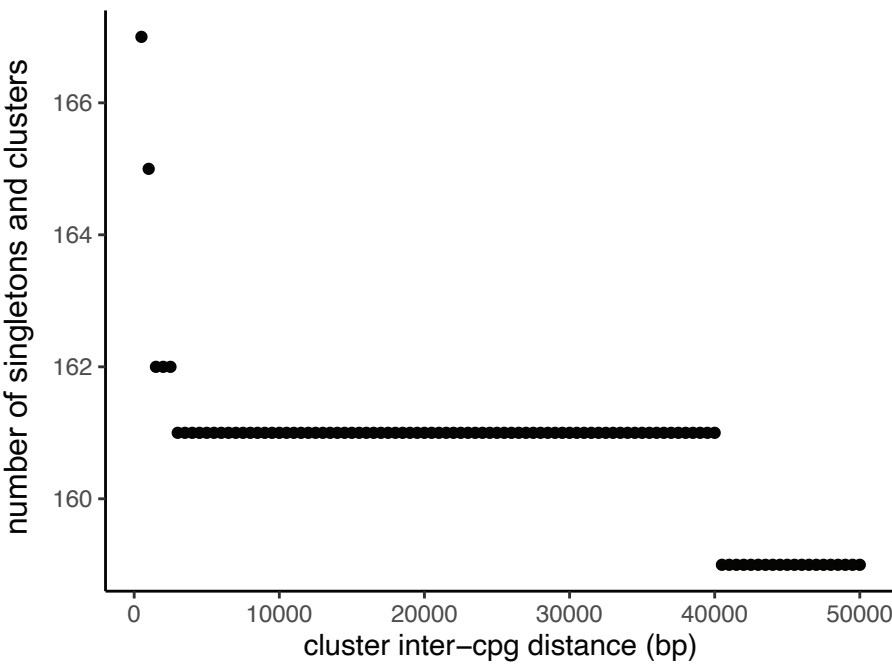

**Appendix 1—figure 5.** Relationship between the inter-CpG distance used to define season of conception (SoC-CpG clusters) and the number of identified clusters and singletons (CpGs not falling within a cluster).

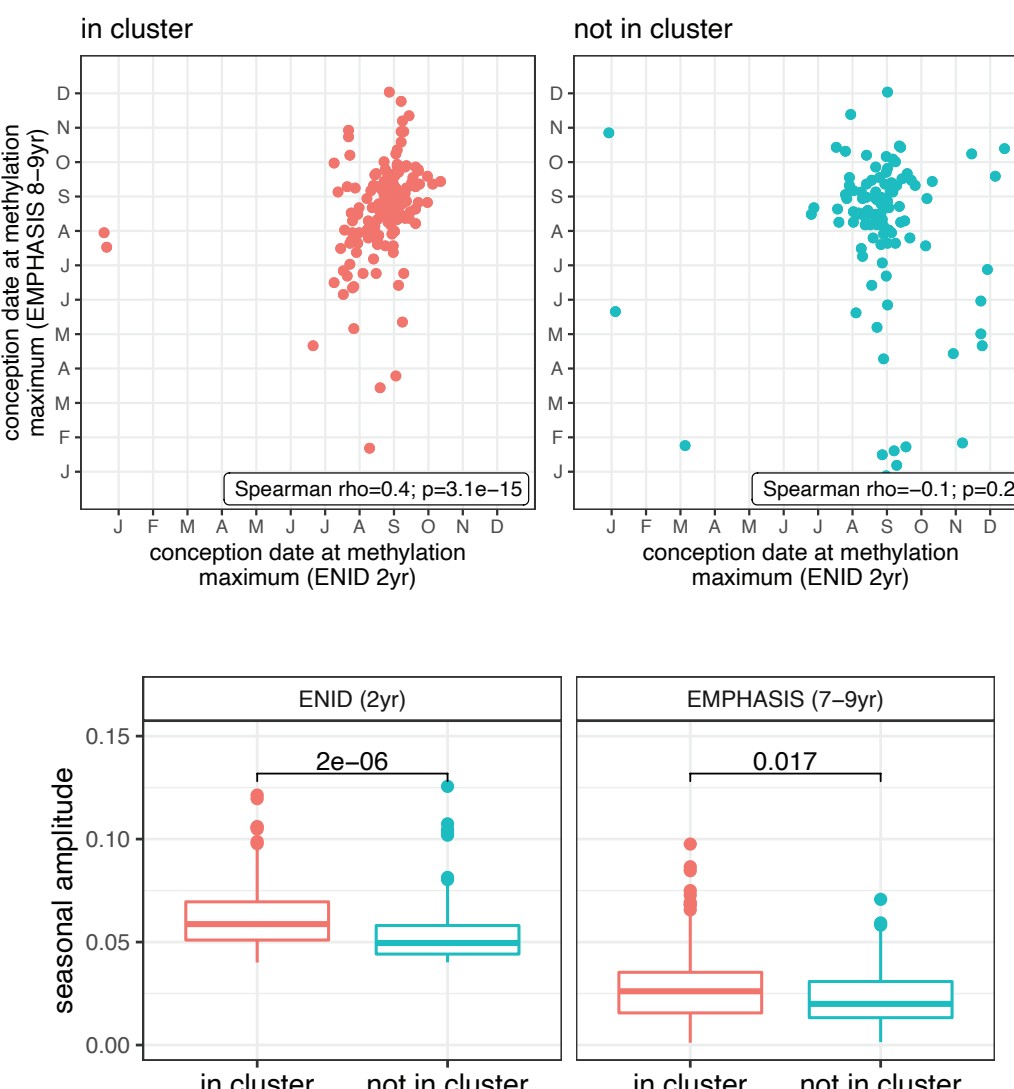

**Appendix 1—figure 6.** Comparison of SoC effects at SoC-CpGs within and outside of SoC-CpG clusters. Top: Conception date at modelled methylation maxima for ENID 2 yr (x-axis) and EMPHASIS (y-axis) cohorts, according to whether locus falls within (*n*=154; left-red), or outside (*n*=105; right-blue) of an SoC-CpG cluster. Bottom: Seasonal effect size (amplitude) for ENID 2 yr (left) and EMPHASIS (cohorts), according to whether locus falls within (red), or outside (blue) of an SoC-CpG cluster. Numbers are p-values for two-sided Wilcoxon rank sum tests under the null hypothesis of no difference between the two distributions.

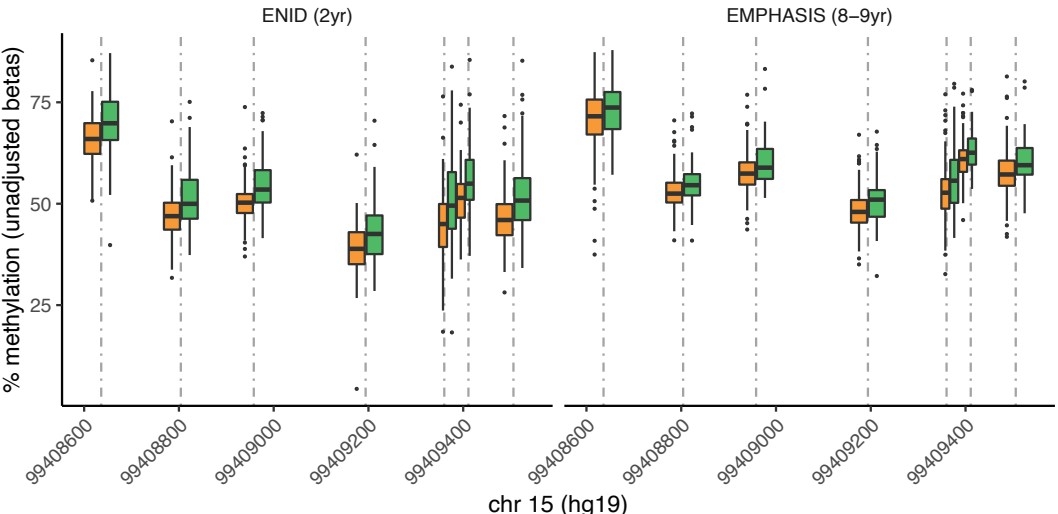

**Appendix 1—figure 7.** Seasonal variation at the IGF1R locus in ENID (2 yr) and EMPHASIS cohorts. Unadjusted methylation beta values are plotted. Boxes represent inter-quartile ranges (IQRs) for season-specific DNAm at each locus. Note that for visualisation purposes Gambian seasons are dichotomised (dry: Jan-Jun; rainy: Jul-Dec), whereas seasonality is modelled as a continuous cyclical variable in Fourier regression models used in the main analysis.

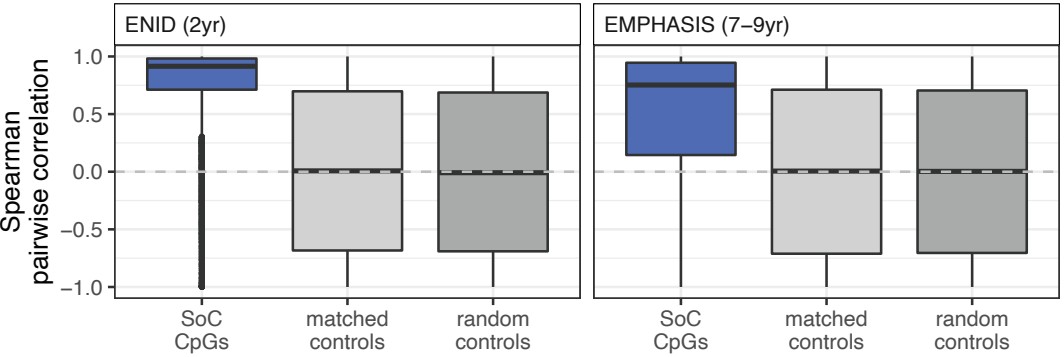

**Appendix 1—figure 8.** Cluster-adjusted pairwise inter-CpG methylation correlations for CpG sets in ENID 2 yr (left) and EMPHASIS (right) cohorts. This is the same as *Figure 4D*, but with a single CpG randomly sampled from each CpG cluster so that CpGs in each set are a minimum distance of 5000 bp apart. This shows that pairwise correlation distributions are not disproportionately driven by large numbers of pairwise correlations between highly correlated neighbours.

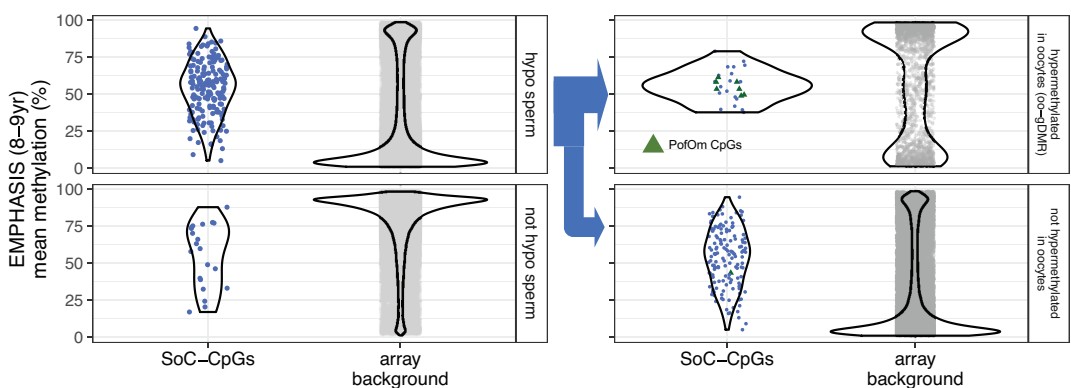

**Appendix 1—figure 9.** Mean methylation at SoC-CpGs and array background measured in *n*=289 individuals in the EMPHASIS (7–9 yr) cohort stratified by sperm and oocyte methylation status. (Left): Methylation stratified according to sperm methylation status reported in **Sugden et al., 2020**. Sperm hypomethylation is defined as methylation ≤ 25%. (Right) As left but with loci hypomethylated in sperm further stratified according to oocyte germline differentially methylated region (oo-gDMR) status reported in **Zink et al., 2018**. Parent-of-origin-specific methylation (PofOm) CpGs are those identified in **James et al., 2018b**. oo-gDMR have oocyte methylation >75%.

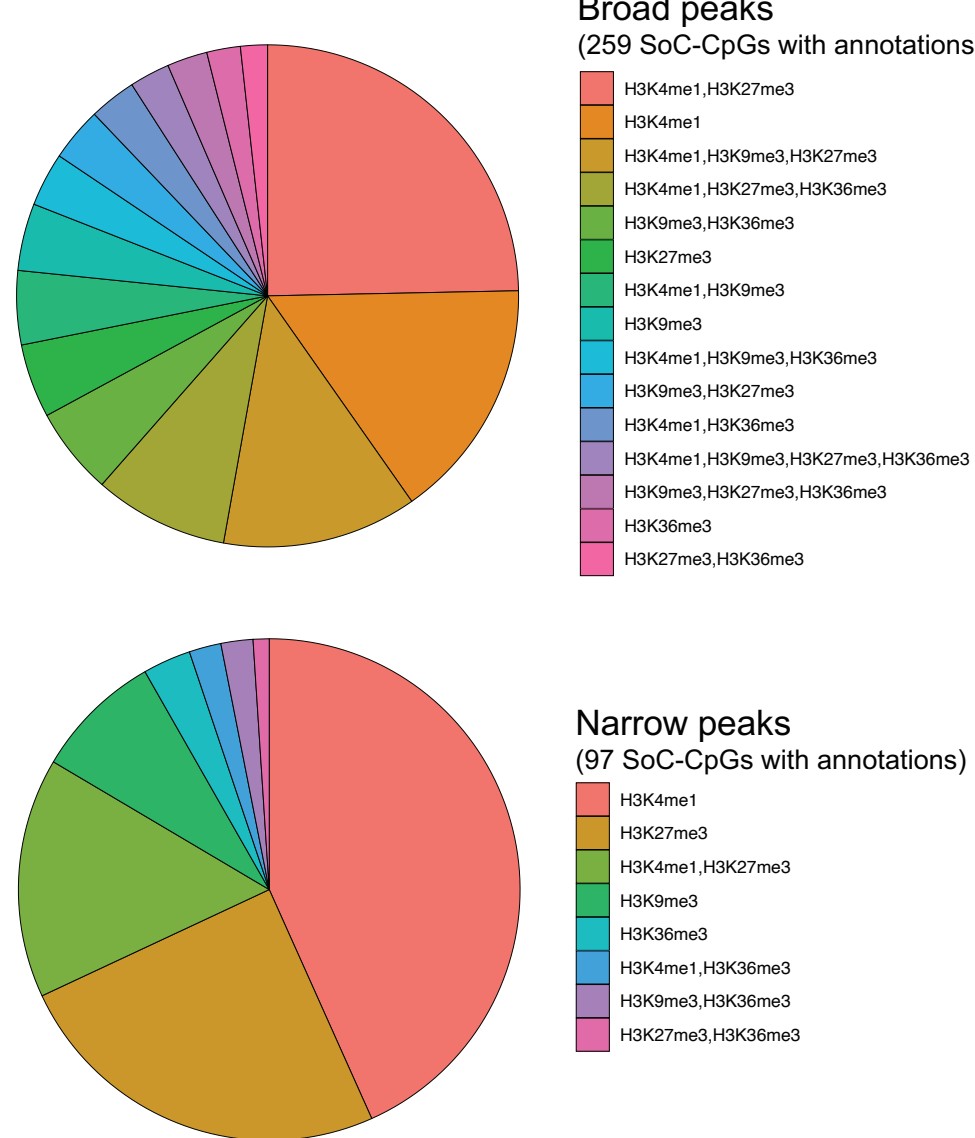

**Broad peaks**
**(259 SoC-CpGs with annotations)**

- H3K4me1,H3K27me3
- H3K4me1
- H3K4me1,H3K9me3,H3K27me3
- H3K4me1,H3K27me3,H3K36me3
- H3K9me3,H3K36me3
- H3K27me3
- H3K4me1,H3K9me3
- H3K9me3
- H3K4me1,H3K9me3,H3K36me3
- H3K9me3,H3K27me3
- H3K4me1,H3K36me3
- H3K4me1,H3K9me3,H3K27me3,H3K36me3
- H3K9me3,H3K27me3,H3K36me3
- H3K36me3
- H3K27me3,H3K36me3

**Narrow peaks**
**(97 SoC-CpGs with annotations)**

- H3K4me1
- H3K27me3
- H3K4me1,H3K27me3
- H3K9me3
- H3K36me3
- H3K4me1,H3K36me3
- H3K9me3,H3K36me3
- H3K27me3,H3K36me3

**Appendix 1—figure 10.** Histone marks overlapping SoC-CpGs in H1 embryonic stem cells. Histone marks or combinations thereof are ordered by abundance. H3 marks were generated by the Roadmap Epigenomics Consortium *Sanchez-Delgado et al., 2016* and downloaded using the annotatr (v1.10.0) package in R. Broad peaks overlapping all 259 SoC-CpGs (top) and (more sharply defined) narrow peaks overlapping 97 SoC-CpGs (bottom) correspond to different thresholds used by the MACS peak-calling algorithm. See *Sanchez-Delgado et al., 2016* for further details.

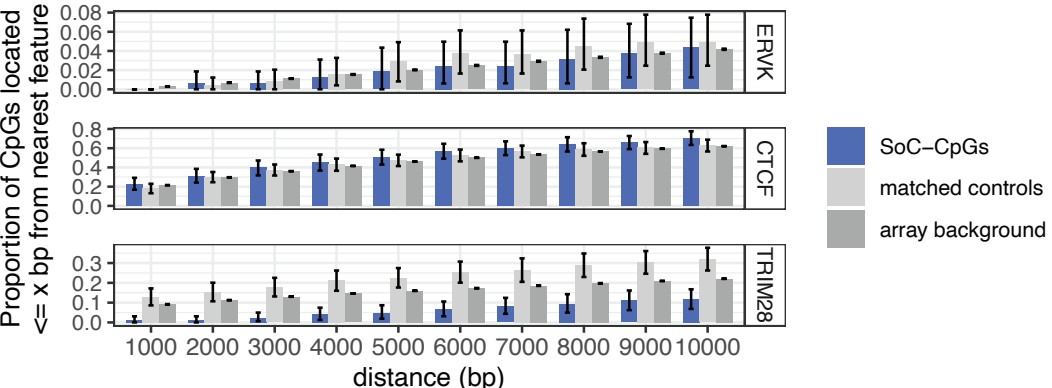

**Appendix 1—figure 11.** Links between ERVK endogenous retroviral elements, ZFP57 and TRIM28 binding sites, and DNAm at SoC-CpGs. (A) Proportion of SoC-CpGs, matched controls, and array background CpGs proximal to ERVK endogenous retroviral elements (top), and ZFP57 and TRIM28 binding sites (bottom), within the specified distance. CpG clustering effects are removed by sampling a single CpG from each cluster (see Materials and methods). Error bars are bootstrapped 95% CIs.

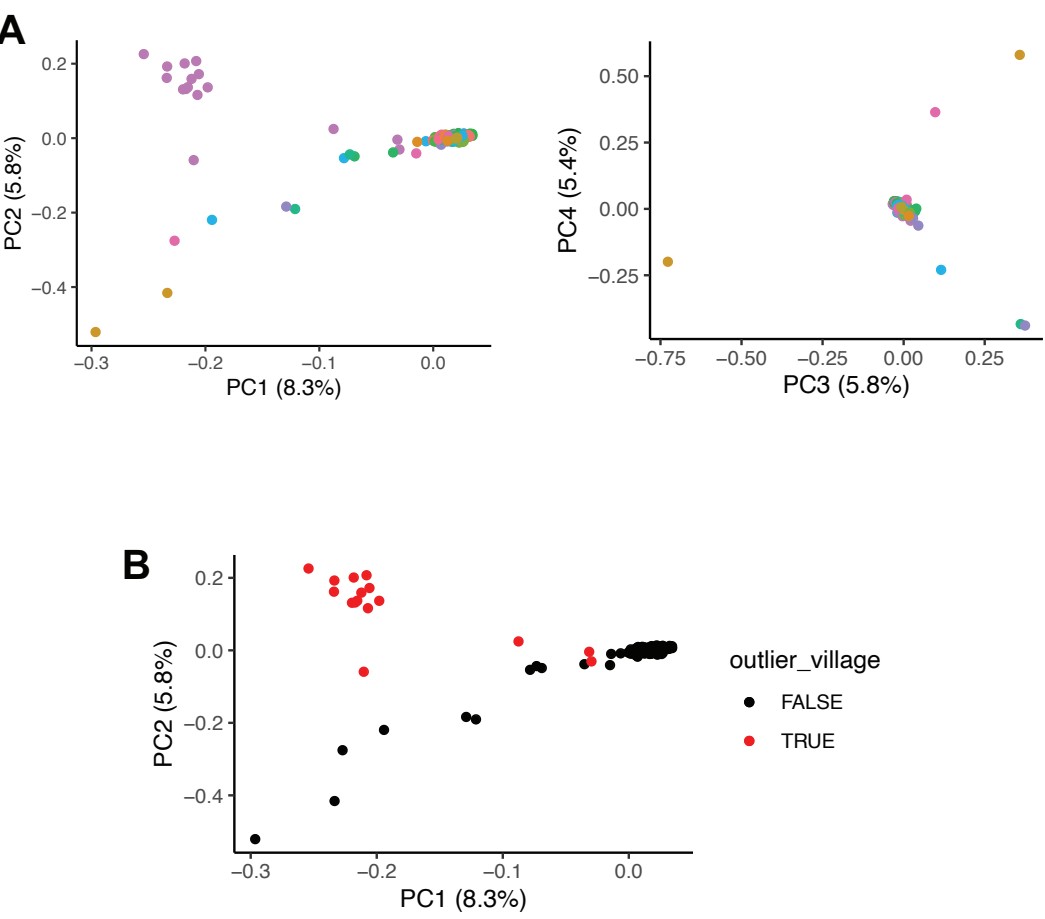

**Appendix 1—figure 12.** Population structure in the EMPHASIS cohort. (A) First and second (left) and second and third (right) principal components (PCs) from a principal component analysis (PCA) of genome-wide genetic data from *n*=294 individuals from the EMPHASIS cohort. Figures in brackets give % variance explained by each PC. Colours correspond to the 24 villages from the Kiang West region which are covered in this cohort. A majority of individuals are of Mandinka origin and villages that are predominantly Mandinka are tightly clustered in the PCA

*Appendix 1—figure 12 continued*

plot. The distinct cluster in the top left of the first plot corresponds to individuals from a single village which is predominantly Fula. (B) Same as A left, but with individuals from the 'outlier village' plotted as a distinct colour. This illustrates that all 16 individuals from this village are distinct from the main cluster.

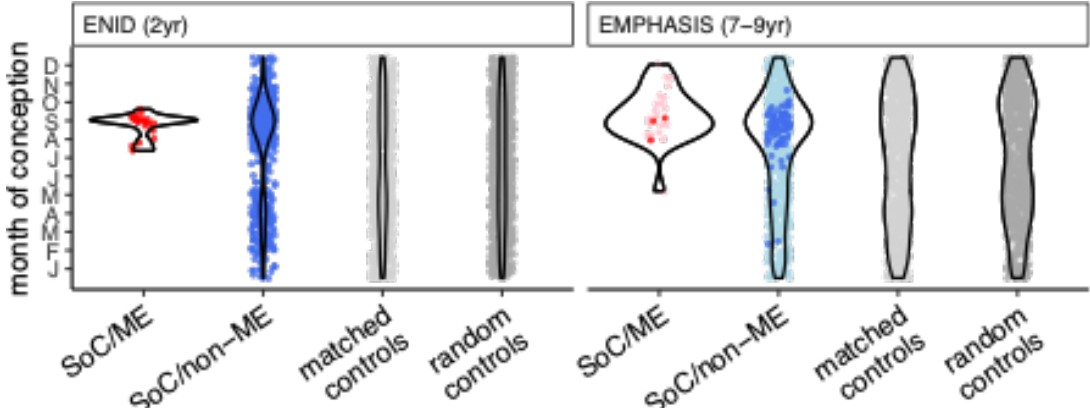

**Appendix 1—figure 13.** Season of conception analysis adjusted for ethnicity. This figure replicates *Figure 4A* in the main paper for the ENID 2 yr and EMPHASIS cohorts, but with additional adjustment for ethnicity. Here, we adjust for ethnicity in the EMPHASIS cohort using the first two genetic principal components as covariates in the Fourier regression models (*Appendix 1—figure 12A*). For the ENID cohort where we do not have genetic data, we adjust for ethnicity using an additional covariate dichotomised according to whether an individual was one of the nine who came from the predominantly Fula 'outlier village' identified from the principal component analysis (PCA) using genetic data in the EMPHASIS cohort (*Appendix 1—figure 12B*).

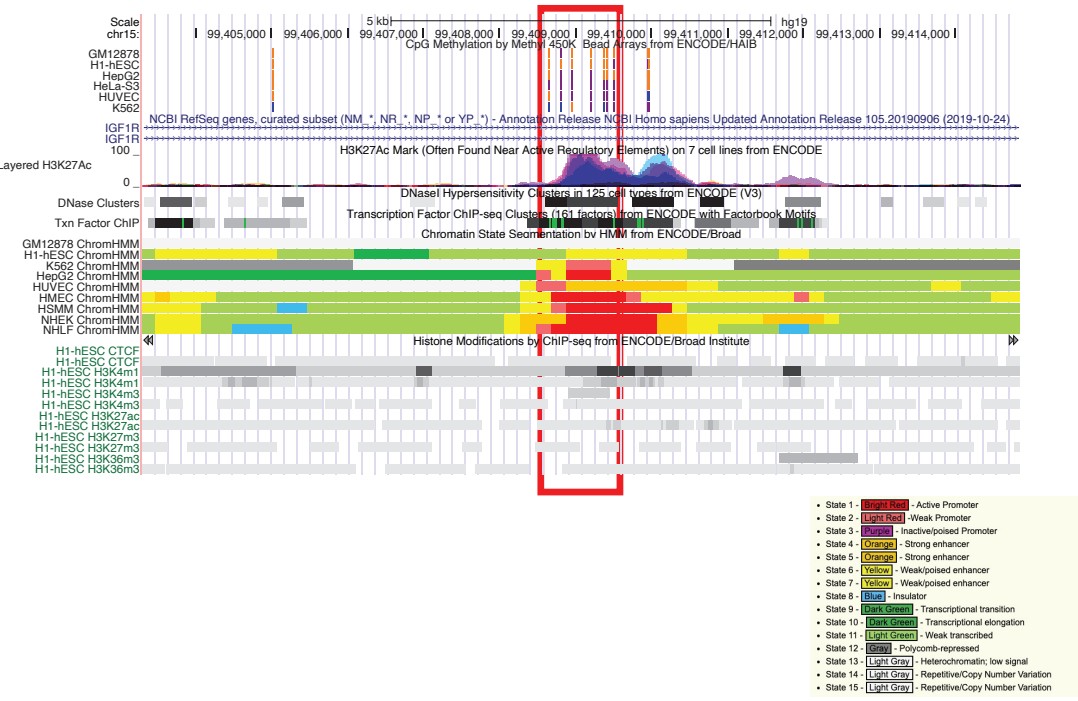

**Appendix 1—figure 14.** UCSC genome browser plot of the IGF1R SoC-CpG cluster. Seven SoC-CpGs on the Illumina 450k array are highlighted. This region falls within intron 2 and bears the hallmarks of being a promoter and/or active or poised enhancer in multiple cell lines. The key lists colour coding for predicted regulatory regions using ChromHMM.

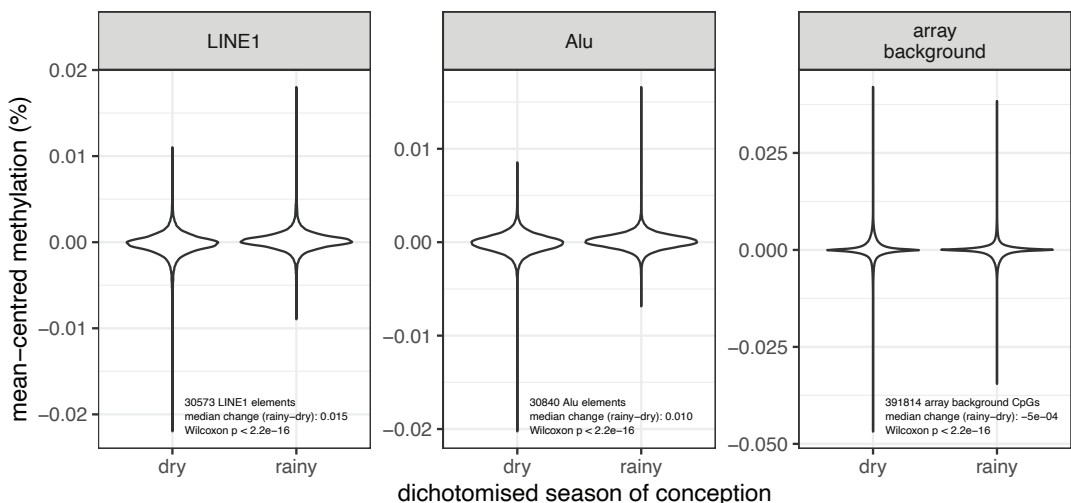

**Appendix 1—figure 15.** Seasonal differences in methylation at 30,573 LINE1 and 30,840 Alu elements, and 391,814 array background CpGs. Methylation values at each locus are centred to have mean zero to enable comparison across loci. LINE1 and Alu methylation values are predicted using REMP (v1.16.0) (see Materials and methods). Season of conception is dichotomised to dry: Jan-June; rainy: July-December.

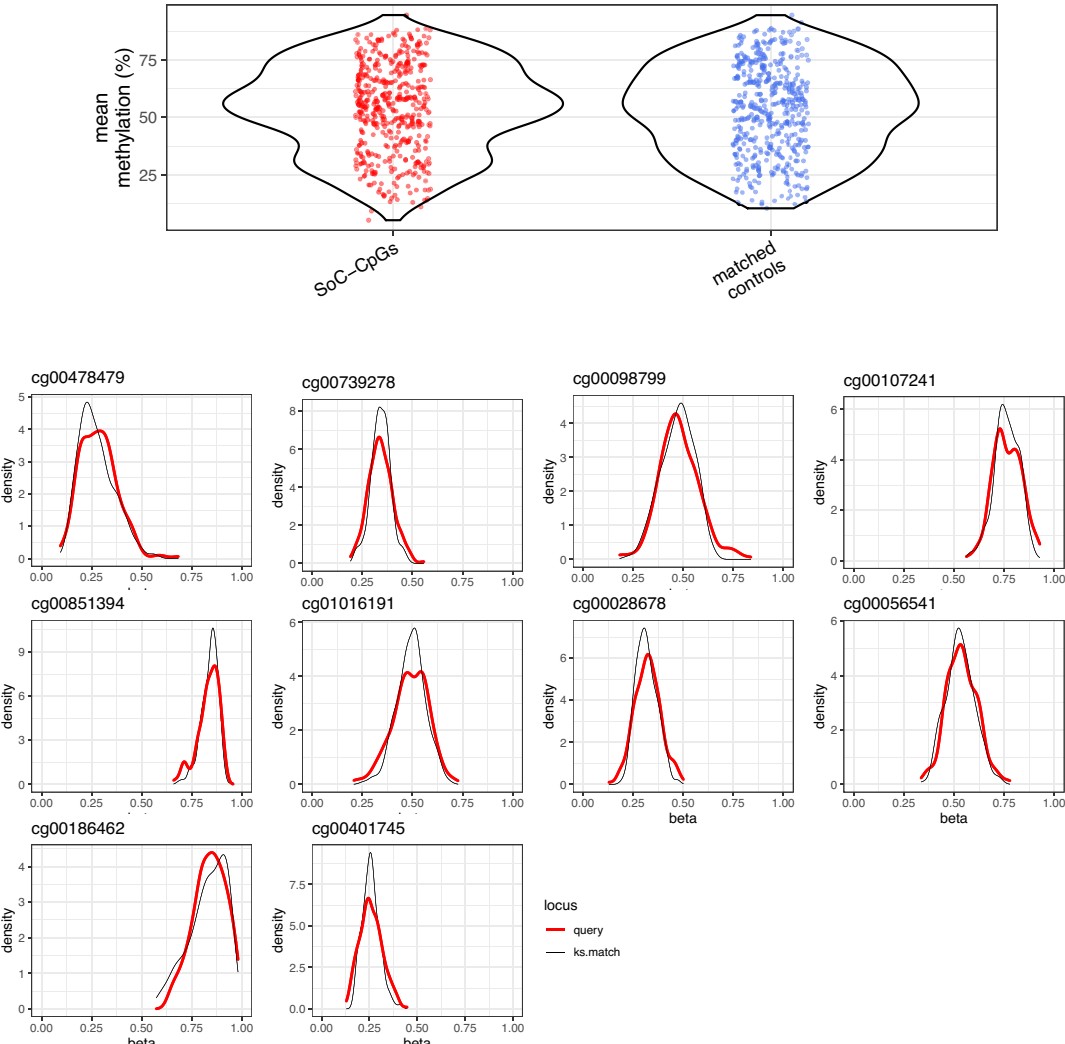

**Appendix 1—figure 16.** Selection of matched controls. Matched controls are selected from array background using Kolmogorov-Smirnov (KS) tests to identify CpGs with similar methylation distributions to SoC-CpGs (see Materials and methods). Top: Methylation mean distribution at 259 SoC-CpGs (left) vs. 259 matched controls (right). Bottom: Methylation distribution of 10 SoC-CpGs selected at random (thick red lines) and their corresponding KS-matched controls (thin black lines).

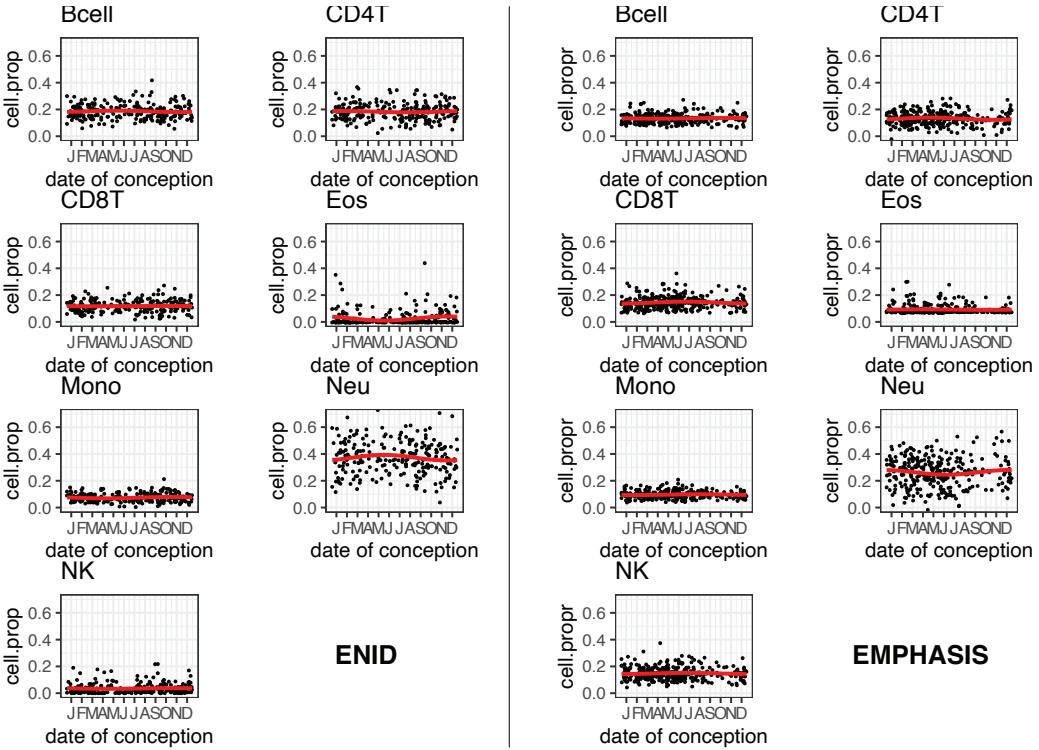

**Appendix 1—figure 17.** Seasonal variation in estimated cell composition. Cell proportions for all samples from each cohort are estimated using the estimateCellCounts function from the minfi package in R. Seasonal variation (red curves) for each cell type is determined from Fourier regression models adjusted for sex (Early Nutrition and Immune Development [ENID]) or age and sex (Epigenetic Mechanisms linking Pre-conceptional nutrition and Health Assessed in India and Sub-Saharan Africa [EMPHASIS]). See Materials and methods for further details.

