## [Editor Report]

This paper investigates the impact of seasonal variation (e.g. nutrition, environment, and infection) in rural subsistence farmer communities in the Gambia on DNA methylation levels in children. The authors identified a set of CpGs that are associated with season of conception and show that these associations are likely driven by periconceptional environmental influences. These findings open the door for future studies of environmentally sensitive CpGs to link early life exposures to diseases occurring later in life.

---

## [Decision Letter]

**Decision letter after peer review:**

Thank you for submitting your article "Environmentally sensitive hotspots in the methylome of the early human embryo" for consideration by *eLife*. Your article has been reviewed by 3 peer reviewers, and the evaluation has been overseen by a Reviewing Editor and Jessica Tyler as the Senior Editor. The following individual involved in review of your submission has agreed to reveal their identity: Toby Mansell (Reviewer #1).

Essential revisions:

Below are first the most important points coming from the discussion between the reviewers, followed by additional points coming from the individual review reports.

1) The reviewers are worried that there is residual confounding impacting the analyses and that the authors only observe increases in DNA methylation. Regressing out sex is not sufficient to show there is no bias in the analyses. We therefore request a correlation matrix (containing rho's, not p-values) containing all batch effects, arrray covariates (e.g. sentrix row and column), biological covariates (e.g. sex), PCAs before and after functional normalisation, and study covariates (e.g. village, month of collection, month of birth, month of conception). This matrix should show if there is residual confounding remaining that should be taken into account in the analyses.

2) The authors should make clear why some decisions were made regarding certain methods (i.e. adjustment using PCA instead of Houseman estimates) and thresholds (i.e. SoC-associated CpGs with difference <4%). More on this in point 14 and 15.

3) The Discussion section should be rewritten to better reflect the relevance of the findings and the limitations of the study. For example, the actual number of unique seasonal MEs is modest and the discussion should reflect this. Moreover, the findings should be compared with previously published studies on, e.g., folic acid supplementation and famine (this last point also applies to the Introduction). More on this in point 12 and 13.

Additional points from individual review reports:

Results section:

4) Page 3, line 94: Have the authors access to longitudinal intra-individual methylation scores during different seasonal windows?

5) Page 4, line 141: Did the authors re-evaluate SoC CpG positions with alternative methods (e.g. pyrosequencing) to analyse larger groups/ cluster of CpG positions and to control the reliability of array data?

6) Line 238 onward: from 257 Soc-CpGs to the "full" set of 768 SoC-CpGs for enrichment tests out of "power" reasons ◊ not consistent and confusing. I would stick with your choice for the 257.

7) Page 7, line 301: The authors mentioned overlaps with previous studies. It would be helpful, if overlapping or related findings could be summarized (and maybe visualized) in more detail. This would facilitate the comparison of the current data set with previous publications.

8) I would strongly recommend reporting the relationship between methylation of these SoC sites and measures of growth or development in these two cohorts – it's not clear to me why this has not been done so, assuming such data is available under approved ethics. In the field there is an ever-growing body of papers linking various early development exposures to differences in methylation in childhood, but there is a dearth of reproduceable evidence for exposures associating with methylation differences that in turn associate with tangible differences in child health/development. Particularly since the manuscript already mentions things like the relationship between methylation and BMI – while adult BMI is different to child measures of body composition, any evidence for a relationship (or a lack of a relationship) with available phenotypes in these two child cohorts would be a logical conclusion that section of analysis and would provide much greater clarity for the relevant Discussion section.

9) Relatedly, if consistent with approved ethics, this paper would benefit from a table summarising the cohort characteristics for each cohort. If appropriate, stratifying the cohorts summaries by season of conception would also be of interest.

10) Rationale for not doing a meta-analysis for power reasons should be explained.

11) The Results section switches from past to present tense and back again. Please keep it to past tense.

Discussion section

12) From my point of view, the observation of an attenuation of DNA methylation levels until mid-childhood is of interest. It would be helpful, if the authors could discuss this point in more detail within the Discussion section. What is known about methylation changes during aging (epigenetic drift etc.) and are these observations relevant for SoC associated CpGs?

13) The authors might discuss in more detail, why these observations of SoC associated loci are disease relevant.

Methods section

14) As mentioned in the public review, I think there needs to be more clarity about why you do and do not use the SoC-associated CpGs with difference <4% – in the results it is stated that the full list is being used to 'maximise power', but it is unclear why maximising power is necessary/better for that analysis compared to the other analyses where only CpGs with difference >4% are used? A little more detail on why it is important/fine to use there would be helpful.

15) Another point that would benefit from additional clarity would be adjusting for cell composition differences in your models – it appears that you have adjusted for these by including the top 6 PCs, but you also have cell count estimates from the Houseman method. Was there are a particular reason for using the PCs when the Houseman estimates were available?

16) Methods does not read as one coherent story, but fragmented and will require some shuffling to make it easier to read.

17) Outlier removal is with 25th and 75th percentile very aggressive compared to what is standard in the field.

18) You can Check SNPs in 450k dataset for possibility to adjust for genetic background in the discovery cohort and 850k cohort to exclude sample swaps/mixtures etc.

19) Reading the methods you are worried by genome-wide effects on DNAm. Why not look at the LINES ALU effects via: Zheng Y Joyce BT Liu L Zhang Z Kibbe WA Zhang W Hou L. Prediction of genome-wide DNA methylation in repetitive elements. Nucleic Acids Res 2017;45:8697-8711 [works for 450k data]

20) Were African allele frequencies used to filter out CpG probes?

21) Was imputation done with AGVP?

22) Was genetic information used to exclude family relationships, mixtures and sample duplications? [knowing the fluidity of family structures and logistics of sample collection in the Gambia]

[Editors' note: further revisions were suggested prior to acceptance, as described below.]

Thank you for resubmitting your work entitled "Environmentally sensitive hotspots in the methylome of the early human embryo" for further consideration by *eLife*. Your revised article has been evaluated by Jessica Tyler (Senior Editor) and a Reviewing Editor.

The manuscript has been improved but there are some remaining issues that need to be addressed, as outlined below. Two of the three original reviewers were happy with the way you addressed their comments, but since the other reviewer was no longer available, I have asked an additional reviewer to specifically take a look at the applied adjustment for batch effects (which was one of the main points brought up by this original reviewer). As you can see below, the new reviewer was in general happy with the rebuttal, but brought up some additional points that need to be addressed.

*Reviewer #4 (Recommendations for the authors):*

Thank you for the opportunity to review the rebuttal to the article titled "Environmentally sensitive hotspots in the methylome of the early human embryo". I would like to say that reading through the rebuttal it appears the authors have been reactive and enthusiastic in their responses to the reviewers. Overall, it appears the changes introduced have help interpretation of the findings as far as I can tell.

In terms of the concerns reviewers have regarding batch effects, on the whole it appears that these have been these addressed as far as is possible within the confines of this particular experiment. The sensitivity analyses reported in table 1s suggest that controlling for the suspected technical confounders (e.g plate, slide, cell proportion estimates) does not significantly attenuate the association between SoC-CpG and doc. As noted, there does appear to be non-randomization with respect to Plate (and by extension, Slide) in the youngest ENID sample, which I suspect is due to assignment of samples to plates as they were collected (i.e in chronological order). What I cannot tell is whether the methylation arrays are also processed as samples are received (i.e. once a plate is filled), or all together at the end of the collection phase. If it is the former, that is a likely source of some variation between plates. Given the use of this cohort as the discovery sample, it would be useful if the authors could declare whether or not this is the case in the methods.

I am however suspicious that there is an alternative confounding factor that could have implications for the interpretation of the results within the paper, and that is the notion of DNA methylation data reliability (or lack of). Multiple studies (e.g. Logue et al. (2017); Epigenomics 9(11):1363-1371; Sugden et al. Patterns 1(2): 100014) have demonstrated that a large proportion of the CpG sites shared between the 450K and EPIC array (as used in this paper) are unreliable, so that it is unlikely they yield the same value when the same sample is measured twice. This of course has implications for determining 'significant associations', for assessing DNA methylation changes over time, and for replication. Because of what we know about reliability and how it manifests, the authors should address reliability in the context of these findings.

Reassuringly for this paper, the SoC-CpGs appear to be pretty robust – cross-referencing the subset of 259 CpGs against the list of reliabilities from Sugden et al., shows the mean ICC is over.7 (close to 'excellent') – this is not the case for the 509 CpGs not taken forward, who have a mean reliability of less than 0.3 ('poor'). The list of CpGs selected as matched and array background controls does not appear available, but I strongly suspect the array background CpGs (at the very least) will have a much lower reliability than the SoC-CpGs (just due to random chance). This clouds the interpretation of any tests that compare these sets of CpGs and should be taken into consideration.

Further, differential reliability should be considered as an alternative explanation for the reduction in mean SoC amplitudes over time – could it be that measurements made using 450K arrays (ENID 2yr) are different to those using EPIC arrays for reasons beyond age-specific effects (i.e. measurement error)?

Finally, the description of the properties of SoC-CpGs (lines 173 onwards) notes that they are more likely to be intermediately methylated – this is again a property of reliable CpGs since arrays do not have the resolution to measure hyper- and hypo- methylated sites very well, leading to unreliable measurements. In all, these points suggest that reliability of DNA methylation is important for interpretation of these results, and, given the high reliability of the SoC-CpGs, is actually a positive feature.

The reviewers were concerned that a large proportion of the discussion was dedicated to MEs and how these findings are related. I believe the authors frame this appropriately, and do not find the references to them excessive. MEs are of great interest, and using cheaper and easier methods to uncover them (such as here) will be beneficial.

---

## [Author Response]

Essential revisions:Below are first the most important points coming from the discussion between the reviewers, followed by additional points coming from the individual review reports.(1) The reviewers are worried that there is residual confounding impacting the analyses and that the authors only observe increases in DNA methylation. Regressing out sex is not sufficient to show there is no bias in the analyses. We therefore request a correlation matrix (containing rho's, not p-values) containing all batch effects, arrray covariates (e.g. sentrix row and column), biological covariates (e.g. sex), PCAs before and after functional normalisation, and study covariates (e.g. village, month of collection, month of birth, month of conception). This matrix should show if there is residual confounding remaining that should be taken into account in the analyses.

We have carried out additional analyses to check for potential residual confounding (see new Supplementary Tables ST1p-1r), including an additional longitudinal analysis with recently acquired methylation data measured on the EPIC array on a subset of individuals from the ENID cohort at age 5-7yr.

Please note that the majority of batch and biological covariates are categorical so that it was not possible to report correlation rho’s. We have instead reported p-values for corresponding association tests – see Supplementary Tables for further details of tests that were carried out. Also note that for simplicity season of conception is modelled as a binary variable (Dry: Jan-Jun; Rainy: July-Dec). We consider this to be a valid approximation to the main cosinor (Fourier) regression analysis since this showed a clear relationship between DNAm and dichotomised (Dry/Rainy) season of conception (Figures2D and 4A). Note that we have not included month of collection as this completely confounds season of conception in the main ENID (2yr) analysis and cannot confound the EMPHASIS (7-9yr) analysis, as discussed in the manuscript (lines 88-91, Figure 2A). This is a key reason why we compared SoC effects across these two cohorts. Note that the month of collection also cannot confound the ENID 5-7yr (longitudinal) analysis as all samples are collected in the rainy season (additional Supp. Figure 2).

The covariate correlation analysis confirms:

– No correlation between SoC and all considered batch and biological covariates including principal components across all three analysed datasets (Supp Table, ST1p-1r).

– No correlation between sex and all considered batch and biological covariates; weak correlations with PC4 and PC3 in EMPHASIS and ENID 5-7yr datasets respectively (ST1q,1r). Please also note also that the sex sensitivity analysis previously reported in the manuscript used methylation values that were *pre-adjusted* for sex using a regression model that included sex as the only adjustment covariate, alleviating concerns that there may be residual confounding due to strong correlations between technical/biological/sampling covariates and sex. We have added some additional comments on this to Results (lines 335-341).

– Expected strong correlations between SoC, month of conception and month of birth in all datasets (ST1p-1r).

– Functional Normalisation (FN) removed most but not all of the effects of technical batch effects (sample plate, slide etc) from the DNAm array data used in the main ENID analysis (ST1p and Supp Table for Reviewers STR1).

– Samples are not perfectly randomised across 450k sample plate (month of birth [mob] and conception [moc]) and slide (mob and village) for the ENID 2yr cohort (ST1p).

The last point raises the possibility of potential residual confounding due to array batch effects in the ENID analysis. We checked for this in two ways. First, we performed sensitivity analyses with batch and village ID variables included directly in the linear regression models, in addition to the PCs that served as proxies for batch variables in our original analysis. This suggested no residual confounding due to array batch or village ID effects (ST1s: ‘batch adjusted model’ and ‘village adjusted model’). Second, we confirmed that neither mob, moc nor village ID were associated with batch or any other covariates in the EMPHASIS or new ENID 5-7yr analyses (ST1q, ST1r). The tight correspondence of date of methylation maximum across all three datasets (cross-cohort and longitudinal analyses) (Figures 2C, 3A and 4A) with different confounding structures (ST1p-1r) strongly suggests that the reported SoC associations are not driven by residual confounding.

In summary, this analysis provides strong reassurance that our main analysis is not confounded by residual associations with technical and/or biological covariates considered in this analysis, and that the observed enrichment for previously identified sex-associations amongst SoC-CpGs is not driven by residual confounding due to sex.

We have made multiple amendments to the manuscript to incorporate the longitudinal analysis; in the Introduction (lines 58-9); in the first section of Results (lines 87-156); and we have made particular reference to the alignment of SoC effects across 3 datasets with different confounding structures in lines 152-156. We have also amended several figure captions to distinguish the ENID 2yr and 5-7yr datasets and added the longitudinal dataset to Methods (lines 558-560) and to the study design schematic (revised Figure 1), and visualised key results from this additional analysis in Figures 3 and 4A. Finally we have added additional text on the sensitivity analyses in the main text (lines 131-133) and in Methods (lines 629-635).

(2) The authors should make clear why some decision were made regarding certain methods (i.e. adjustment using PCA instead of Houseman estimates) and thresholds (i.e. SoC-associated CpGs with difference <4%). More on this in point 14 and 15.

Regarding adjustment for cell count estimates using PCs as a proxy, we have performed an additional sensitivity analysis with Houseman estimated blood cell counts added directly to the linear regression model for the ENID cohort (see ST1s). 518 out of the 520 estimated Fourier regression coefficients from the main analysis (1 pair of sine and cosine terms for each of the 259 SoC-CpGs) fall within the 95% confidence interval obtained in the Houseman-adjusted analysis, confirming that cell composition effects did not unduly influence SoC effect estimates in the original analysis. As outlined in our response to the previous comment we have added a brief note on this and the other sensitivity analyses (batch, cell composition and village effects) in Results to the manuscript (lines 131-133), with more details in Methods (lines 629-635).

Regarding our use of different SoC amplitude thresholds for one analysis, our original motivation for analysing all 768 ‘SoC-associated CpGs’ with FDR<5% in the ENID 2yr analysis, including those with amplitude < 4%, was to explore the degree to which the strength / amplitude of SoC effects could be explained by proximity to ERV1 over the wider range of amplitudes represented by the larger set of loci. However we agree that this approach is open to question and have removed this analysis (previous Figure 6B and Supp. Figure 11, and text in section headed ‘Enrichment of transposable elements and transcription factors associated with genomic imprinting’). We have also removed the definition of ‘SoC-associated CpGs’ (which included CpGs with SoC amplitude < 4%) from Table 2 and Methods to aid clarity and avoid confusion.

(3) The Discussion section should be rewritten to better reflect the relevance of the findings and the limitations of the study. For example, the actual number of unique seasonal MEs is modest and the discussion should reflect this.

This reviewer (see additional comments in the public evaluation summary) acknowledges that the fold-enrichment for previously identified MEs is large, but they are concerned that MEs are the main focus of the Results and Discussion and are left wondering what could have been found in the large majority (approx. 90%) of SoC-CpGs that do not overlap known MEs.

Our strategy throughout the Results and Discussion was to focus on characteristics including metastability, parent of origin-specific methylation, histone modifications and gametic and early embryo methylation patterns that suggest a link to establishment of methylation states in the early embryo at SoC-CpGs. For these analyses *all* SoC-CpGs were considered at every stage and metastability was not the primary focus. However, we do repeatedly point out that many of the contextual genomic features associated with SoC-CpGs have also been associated with metastability which we consider to be worthy of note, in part because it suggests that many SoC-CpGs may in fact be MEs, despite not having been previously identified as such. We have further cause to believe this could be the case because of (i) the typically small sample size of multi-germ layer/multi-tissue datasets used in previous screens for MEs, meaning that published screens for human MEs are underpowered and will hence fail to capture most MEs; and (ii) the evidence that we present suggesting that environmentally-driven inter-individual variation at loci exhibiting ME-like properties may diminish with age, again suggesting that ME screens, which tend to analyse adult tissues, will miss metastable loci present in infancy and early childhood.

We had already made the point (ii) above in the Discussion (lines 383-389. However, given the reviewer’s concerns we have added an additional comment on point (i)) (lines 389-392).

Moreover, the findings should be compared with previously published studies on, e.g., folic acid supplementation and famine (this last point also applies to the Introduction). More on this in point 12 and 13.

We did in fact compare our results with previously published studies looking at both famine and folic acid supplementation that were included in a recent review of DNAm loci associated with maternal nutrient exposures (James et al. IJE 2018; see section titled ‘Overlap of SoC-CpGs with existing studies’). For ease of reference we have included an additional table summarising overlaps with loci highlighted in the James et al. review (ST1k). We have also added an explicit reference to the fact that folate supplementation was considered in this comparison to the main text (line 310) and have also cited existing evidence on links to periconceptional folate and famine in the Introduction (lines 50-51).

Additional points from individual review reports:Results section:(4) Page 3, line 94: Have the authors access to longitudinal intra-individual methylation scores during different seasonal windows?

As outlined above, we are pleased to report that we have now generated additional EPIC array data covering a subset of n=138 individuals from the ENID cohort included in the main analysis. This subset had methylation measured in blood at age 5-7yrs enabling us to conduct an investigation of longitudinal methylation changes in these individuals. This analysis strongly supports the evidence of SoC effect attenuation with age suggested by our previous comparison of the independent ENID (2yr) and EMPHASIS (7-9yr) cohorts, with:

(a) Strong correlation of conception date methylation maximum between age 2yrs and 5-7yrs at SoC-CpGs in these 138 individuals (Figures 3A, 4A); and

(b) Evidence of SoC effect size attenuation at the majority of SoC-CpGs (Figure 3B; Wilcoxon signed rank sum p=10^-12^).

Note that as described above, this additional longitudinal dataset also has a different confounding structure with respect to biological and technical covariates (Supp Tables 1p-1r) and date of sample collection (Supp. Figure 2), strongly supporting our previous analysis.

Interestingly, a larger number of SoC-CpGs replicate in this smaller and slightly younger longitudinal dataset, suggesting a possible secular or cohort effect. We have added some additional comments on this to the Discussion (lines 400-403) and have highlighted the different years of conception between the two analysed cohorts in Methods (lines 554,563).

(5) Page 4, line 141: Did the authors re-evaluate SoC CpG positions with alternative methods (e.g. pyrosequencing) to analyse larger groups/ cluster of CpG positions and to control the reliability of array data?

We did not perform a technical validation using an alternative DNAm measurement method and unfortunately DNA stocks from ENID 2yr individuals studied in the main analysis are now exhausted. However we consider the strong concordance of our findings, which with the additional dataset now span 3 datasets with different confounding structures, to be a strong technical validation of our findings.

(6) Line 238 onward: from 257 Soc-CpGs to the "full" set of 768 SoC-CpGs for enrichment tests out of "power" reasons ◊ not consistent and confusing. I would stick with your choice for the 257.

Please see our response to point (2) above.

(7) Page 7, line 301: The authors mentioned overlaps with previous studies. It would be helpful, if overlapping or related findings could be summarized (and maybe visualized) in more detail. This would facilitate the comparison of the current data set with previous publications.

As outlined in a previous response above, we have listed regions associated with periconception and gestational exposures covered in the cited review (James et al. IJE, 2018) that overlap the array background analysed in our study in a new Supplementary Table (Supp. Table 1k).

(8) I would strongly recommend reporting the relationship between methylation of these SoC sites and measures of growth or development in these two cohorts – it's not clear to me why this has not been done so, assuming such data is available under approved ethics. In the field there is an ever-growing body of papers linking various early development exposures to differences in methylation in childhood, but there is a dearth of reproduceable evidence for exposures associating with methylation differences that in turn associate with tangible differences in child health/development. Particularly since the manuscript already mentions things like the relationship between methylation and BMI – while adult BMI is different to child measures of body composition, any evidence for a relationship (or a lack of a relationship) with available phenotypes in these two child cohorts would be a logical conclusion that section of analysis and would provide much greater clarity for the relevant Discussion section.

We are currently researching links between several SoC-CpGs and health-related outcomes including measures of growth, and we have prepared/submitted other papers with different groups of authors (e.g. the EMPHASIS team) relating to other phenotypes. We consider a detailed analysis of links between SoC-CpGs and diverse outcome measures in Gambian children to be beyond the scope of the current study and would argue that such an analysis would dilute the central focus of this paper that is already long and complex. In the current paper we do already refer to two existing studies linking Gambian SoC or nutrition-associated CpGs to health outcomes in non-Gambians (child and adult obesity/POMC, Kuhnen et al. Cell Metab 2016; cancer/VTRNA2-1, Silver et al., Gen Biol 2015) in the current manuscript (lines 513-515 and 419-422). The VTRNA2-1 locus does not overlap any SoC-CpGs and we already speculate that this may be due to SoC effect attenuation, since the previous association was observed in younger (3-9mth) infants (lines 443-445). We have additionally referenced a recently published paper linking another SoC-associated locus to thyroid volume and function in Gambian children (Candler et al. Sci Adv 2021; lines 515-522) and highlighted that neither this nor the *POMC* locus overlap the array background analysed in this study. Finally we have already included an analysis of overlaps between SoC-CpGs and traits in published EWAS and GWAS catalogues (lines 320-347).

(9) Relatedly, if consistent with approved ethics, this paper would benefit from a table summarising the cohort characteristics for each cohort. If appropriate, stratifying the cohorts summaries by season of conception would also be of interest.

As above, we think that inclusion of additional phenotypic characteristics for individuals with DNAm measures covering the two cohorts (and two timepoints) analysed is beyond the scope of this study. We strongly prefer to focus on the robust exposure – methylation associations across cohorts and timepoints identified in this paper.

(10) Rationale for not doing a meta-analysis for power reasons should be explained.

The rationale for not performing a meta-analysis is our observation, now supported by longitudinal data, that SoC effect sizes diminish with age. This means that a combined meta-analysis of all datasets would have reduced power to detect SoC-CpGs present at age 2yr. We have added a note to this effect in Methods (lines 625-628).

(11) The Results section switches from past to present tense and back again. Please keep it to past tense.

If the editor agrees we would prefer to stick to the current style of referring to analyses performed in the past tense (“we investigated…”, “we analysed…”, “we tested…”), with key findings in the present tense (”SoC-CpGs are distributed throughout the genome…”). The manuscript has been read by a large number of authors, colleagues and reviewers and this point has not been raised by anyone except this reviewer.

Discussion section(12) From my point of view, the observation of an attenuation of DNA methylation levels until mid-childhood is of interest. It would be helpful, if the authors could discuss this point in more detail within the Discussion section. What is known about methylation changes during aging (epigenetic drift etc.) and are these observations relevant for SoC associated CpGs?

We have added some further comment on this in the Discussion (lines 397-399).

(13) The authors might discuss in more detail, why these observations of SoC associated loci are disease relevant.

Apart from our own studies (see response to point 8 above), Gunasakera et al. (Epigenomics 11, 2019) note that stochastic and/or environmentally influenced SIV (systemic interindividual variation) loci identified by our group and others have been associated with a number of diseases including Alzheimer’s, cancer, rheumatoid arthritis and schizophrenia. We have added some further comment on this to the Discussion (lines 520-522).

Methods section(14) As mentioned in the public review, I think there needs to be more clarity about why you do and do not use the SoC-associated CpGs with difference <4% – in the results it is stated that the full list is being used to 'maximise power', but it is unclear why maximising power is necessary/better for that analysis compared to the other analyses where only CpGs with difference >4% are used? A little more detail on why it is important/fine to use there would be helpful.

We have now removed the ERV1 proximity vs amplitude analysis that included sub-threshold SoC-associated CpGs. See our response to (2) above for further details.

15) Another point that would benefit from additional clarity would be adjusting for cell composition differences in your models – it appears that you have adjusted for these by including the top 6 PCs, but you also have cell count estimates from the Houseman method. Was there are a particular reason for using the PCs when the Houseman estimates were available?

We have performed an additional sensitivity analysis to address this point. See our response to (2) above.

(16) Methods does not read as one coherent story, but fragmented and will require some shuffling to make it easier to read.

We have reviewed Methods and included additional material as outlined in our response to comments above. We hope it is now easier to follow.

(17) Outlier removal is with 25th and 75th percentile very aggressive compared to what is standard in the field.

In fact we used a common, much less aggressive threshold, sometimes referred to as ‘Tukey’s method’, with outliers defined as 3 x the inter-quartile range (IQR) in either direction (see Methods lines 621-624).

(18) You can Check SNPs in 450k dataset for possibility to adjust for genetic background in the discovery cohort and 850k cohort to exclude sample swaps/mixtures etc.

We acknowledge the potential for further refinement of our analysis using genotypes imputed from the methylation data. However we believe such an analysis is highly unlikely to alter the main findings. In particular we note (a) the strong evidence that major genetic PCA clusters are driven by ethnicity (Mandinka vs Fula; Supp. Figure 12), with village serving as a strong proxy for this; and (b) the strong concordance of SoC effects between ENID 2yr and EMPHASIS 7-9yr cohorts in the sensitivity analysis where we adjust for genetic background in the EMPHASIS cohort, and for village as a proxy for ethnicity in the ENID 2yr cohort (main text lines 293-307; Methods lines 782-789; Supp. Figure 13; sensitivity analysis described in ST1s). Finally, again given the strong concordance of SoC effects between the two cohorts described above, we think it highly improbable that sample swaps or mixtures revealed by an analysis of methylation-imputed genotypes in the EMPHASIS cohort would materially alter our main findings.

(19) Reading the methods you are worried by genome-wide effects on DNAm. Why not look at the LINES ALU effects via: Zheng Y Joyce BT Liu L Zhang Z Kibbe WA Zhang W Hou L. Prediction of genome-wide DNA methylation in repetitive elements. Nucleic Acids Res 2017;45:8697-8711 [works for 450k data]

Thank you for this suggestion. We have now analysed potential SoC effects on global methylation at LINE1 and Alu elements using the suggested method published by Zheng et al. This confirmed a small but significant effect of increased methylation in rainy vs dry conceptions at both LINE1 and Alu elements (0.02% and 0.01% respectively, both Wilcoxon p<2.2x10^-16^) suggesting a SoC effect on global methylation levels. This contrasts with a significant but extremely small effect in the opposite direction across array background (-5x10^-4^%; p<2.2x10^-16^). This supports evidence of a similar effect of peri/pre-conceptional folate (a one-carbon metabolite that varies significantly by season in our population; Dominguez-Salas et al. Nature Communications, 2014) on global methylation levels observed in previous studies.

We have updated the section on inflation in Methods to include this additional analysis (lines 647-655) and summarised the findings from the LINE1/ALU analysis in Supp. Figure 15.

(20) Were African allele frequencies used to filter out CpG probes?

Probes in or near SNPs were masked using the Africa-specific (GWD) list provided by Zhou et al., Comprehensive characterization, annotation and innovative use of Infinium DNA methylation BeadChip probes. Nucleic Acids Research 45, e22 (2017).

(21) Was imputation done with AGVP?

As outlined in Methods (lines 736-738), imputation was performed using 1000 Genomes phase 3 data as a reference.

(22) Was genetic information used to exclude family relationships, mixtures and sample duplications? [knowing the fluidity of family structures and logistics of sample collection in the Gambia]

See response to (18) above regarding mixtures/duplications. Regarding family relationships, there is indeed a relatively high frequency of half-sibs in our study population. We were unable to test this directly in the discovery (ENID 2yr) cohort as genetic data were not available. However, of the 199 ENID children with paternal ID data we identified 2 pairs of half-sibs each with a common father, corresponding to 2% of participants. This number is too small to influence our findings. We have added a note on this in the Discussion (lines 534-537).

Finally, please note that the age range of the EMPHASIS cohort was previously incorrectly written as 8-9yrs throughout the manuscript. This has been changed to 7-9yrs to reflect the fact that 6 out of the 233 individuals in this cohort were aged 7.

[Editors' note: further revisions were suggested prior to acceptance, as described below.]

Reviewer #4 (Recommendations for the authors):Thank you for the opportunity to review the rebuttal to the article titled "Environmentally sensitive hotspots in the methylome of the early human embryo". I would like to say that reading through the rebuttal it appears the authors have been reactive and enthusiastic in their responses to the reviewers. Overall, it appears the changes introduced have help interpretation of the findings as far as I can tell.In terms of the concerns reviewers have regarding batch effects, on the whole it appears that these have been these addressed as far as is possible within the confines of this particular experiment. The sensitivity analyses reported in table 1s suggest that controlling for the suspected technical confounders (e.g plate, slide, cell proportion estimates) does not significantly attenuate the association between SoC-CpG and doc.As noted, there does appear to be non-randomization with respect to Plate (and by extension, Slide) in the youngest ENID sample, which I suspect is due to assignment of samples to plates as they were collected (i.e in chronological order). What I cannot tell is whether the methylation arrays are also processed as samples are received (i.e. once a plate is filled), or all together at the end of the collection phase. If it is the former, that is a likely source of some variation between plates. Given the use of this cohort as the discovery sample, it would be useful if the authors could declare whether or not this is the case in the methods.

We can confirm that the samples were processed in two batches. Samples were not processed sequentially by date of collection and each processing batch covered conception dates throughout the year. We have added this information to Methods (lines 578-80):

“Samples from the ENID 2yr dataset were processed in two batches with each batch covering conception dates throughout the year. Samples from the ENID (5-7yr) and EMHASIS (7-9yr) datasets were processed in single batches.”

As the reviewer notes we previously took additional steps to check for the possibility of residual confounding due to batch/technical effects including carrying out sensitivity analyses with batch effects included directly in the model. We have additionally checked and confirmed that adding processing batch to these models had no significant effect on estimated Fourier regression coefficients in the main SoC association analysis.

Importantly, as previously noted we also observed a tight correspondence of date of methylation maximum across all three datasets considered (Figures 2C, 3A and 4A). Two of these datasets (longitudinal ENID dataset with DNAm measured at 5-7yrs and EMPHASIS dataset measured at 7-9yrs) had very different confounding structures with no associations between date of conception / birth / collection and batch/technical variables.

I am however suspicious that there is an alternative confounding factor that could have implications for the interpretation of the results within the paper, and that is the notion of DNA methylation data reliability (or lack of). Multiple studies (e.g. Logue et al. (2017); Epigenomics 9(11):1363-1371; Sugden et al. Patterns 1(2): 100014) have demonstrated that a large proportion of the CpG sites shared between the 450K and EPIC array (as used in this paper) are unreliable, so that it is unlikely they yield the same value when the same sample is measured twice. This of course has implications for determining 'significant associations', for assessing DNA methylation changes over time, and for replication. Because of what we know about reliability and how it manifests, the authors should address reliability in the context of these findings.

We agree that probe reliability is an important consideration, although we note that this is likely to increase the number of false negatives, since low reliability probes would be unlikely to be picked up in our discovery analysis, and methylation maxima would be unlikely to align across datasets.

Reassuringly for this paper, the SoC-CpGs appear to be pretty robust – cross-referencing the subset of 259 CpGs against the list of reliabilities from Sugden et al., shows the mean ICC is over.7 (close to 'excellent') – this is not the case for the 509 CpGs not taken forward, who have a mean reliability of less than 0.3 ('poor'). The list of CpGs selected as matched and array background controls does not appear available, but I strongly suspect the array background CpGs (at the very least) will have a much lower reliability than the SoC-CpGs (just due to random chance). This clouds the interpretation of any tests that compare these sets of CpGs and should be taken into consideration.

We confirm that the probe reliability as measured by Sugden et al. is ‘excellent’ for SoC-CpGs (median ICC = 0.76). The same measure for distribution-matched control CpGs is 0.68 (‘good’), providing reassurance that analysed DNAm values for both sets of probes are reliable. This aligns with observations by Xu and Taylor (Epigenetics, 2021 May;16(5):495-502) that low ICC scores are associated with sites with low variability.

We agree that array background and random control probes will have much lower reliability, as assessed by Sugden et al. In almost all cases for the present study, analyses that include random controls or array background also included a comparison with matched controls. In these cases we think it is still useful to include random controls and/or array background as it gives the reader useful information on how our analyses relate to properties of the 450k array. To take two examples, it is instructive to see 450k DNAm distributions in the embryonic inner cell mass and embryonic liver (Figure 4A); and also to see how 450k probes are distributed with respect to ERVs and ZFP57 binding sites (Figure 6A). There is one example where SoC-CpGs are compared with array background only (Figure 4D). Again we think the comparison is justified as it highlights array-wide associations between sperm and oocyte hypo/hyper methylation and hypo/hyper methylation in blood, rather than making any detailed quantitative comparisons.

We have added the following to Results (lines 184-186):

“SoC-CpG reliability is classified as ‘excellent’ (median ICC=0.76) using probe reliability estimates from a recent repeated measures study^23^. Reliability of matched control CpGs is classified as ‘good’ (median ICC=0.68) using the same method.”

We have also added a comment on the issue of general unreliability of probes overlapping the 450k and EPIC arrays to the Discussion, including the additional point on intermediate methylation and probe reliability raised by the reviewer below (lines 535-538):

“Measured DNAm values at SoC-CpGs and matched controls have previously been reported to be reliable, in strong contrast to the majority of probes overlapping the 450k and EPIC arrays that have been found to have low test-retest reliability^23^. This likely reflects increased inter-individual variability and/or intermediate DNAm at SoC-CpGs and controls^75,76^.”

23: Sugden et al. Patterns (2020); 75: Logue et al. Epigenomics (2017); 76: Xu and Taylor Epigenetics (2021).

Further, differential reliability should be considered as an alternative explanation for the reduction in mean SoC amplitudes over time – could it be that measurements made using 450K arrays (ENID 2yr) are different to those using EPIC arrays for reasons beyond age-specific effects (i.e. measurement error)?

We think this is unlikely given the high reliability of these probes discussed above, and given the strong concordance between DNAm maxima further suggesting consistency between arrays.

Finally, the description of the properties of SoC-CpGs (lines 173 onwards) notes that they are more likely to be intermediately methylated – this is again a property of reliable CpGs since arrays do not have the resolution to measure hyper- and hypo- methylated sites very well, leading to unreliable measurements. In all, these points suggest that reliability of DNA methylation is important for interpretation of these results, and, given the high reliability of the SoC-CpGs, is actually a positive feature.

Thank you. We agree that increased reliability of intermediately methylated probes is reassuring. We have added the point about increased reliability of intermediately methylated probes – see added text above.

The reviewers were concerned that a large proportion of the discussion was dedicated to MEs and how these findings are related. I believe the authors frame this appropriately, and do not find the references to them excessive. MEs are of great interest, and using cheaper and easier methods to uncover them (such as here) will be beneficial.

We thank the reviewer for their comment which we certainly agree with.